# Modulating the PPARγ pathway upregulates NECTIN4 and enhances chimeric antigen receptor (CAR) T cell therapy in bladder cancer

Kevin Chang[1,2,3], Henry M. Delavan[1,2], Elizabeth Yip[1,2,13], Corynn Kasap[1,2,4], Jun Zhu[1,5], Roshan Lodha [1,2], Sheng-You Liao[6], Sarah C. Berman[1,2], Alberto Carretero-Gonzalez [1,2], Merve Basar[7], Gamze Gokturk Ozcan[7], Min Yuen Teo [8], David B. Solit [8,9], Jonathan E. Rosenberg[8], Hikmat Al-Ahmadie [7], Cornelia C. K. Ding [1,10], Emily Chan[1,10,11], Veronica Steri[1], Sima P. Porten[1,3], Vadim S. Koshkin [1,2], Terence W. Friedlander[1,2], Felix Y. Feng [1,2,3,5,12], John K. Lee [6], Arun P. Wiita [1,4], Carissa E. Chu[1,3,9] & Jonathan Chou [1,2] ✉

With the approval of the antibody-drug conjugate enfortumab vedotin (EV), NECTIN4 has emerged as a bona fide therapeutic target in urothelial carcinoma (UC). Here, we report the development of a NECTIN4-directed chimeric antigen receptor (CAR) T cell, which exhibits reactivity across cells expressing a range of endogenous NECTIN4, with enhanced activity in high expressors. We demonstrate that the PPARγ pathway, critical for luminal differentiation, transcriptionally controls *NECTIN4*, and that the PPARγ agonist rosiglitazone primes and augments NECTIN4 expression, thereby increasing sensitivity to NECTIN4-CAR T cell-mediated killing. NECTIN4-CAR T cells have potent antitumor activity even against EV resistant cells, which largely retain NECTIN4 expression, including in a post-EV biopsy cohort. Our results elucidate a therapeutically actionable mechanism that UC cells use to control NECTIN4 expression and suggest therapeutic approaches that leverage PPARγ agonists for rational combinations with NECTIN4-targeting agents in UC, as well as future potential treatment options for EV-refractory patients.

Urothelial carcinoma (UC) is the second most common genitourinary malignancy, leading to over 16,000 deaths per year in the United States[1]. In spite of recent advances, patients who develop metastatic UC (mUC) have a 5-year survival of ~5%. Enfortumab vedotin (EV) is the first FDA-approved ADC targeting NECTIN4, an immunoglobulin-like, transmembrane protein which is highly and heterogeneously expressed in UC, validating NECTIN4 as a bona fide target[2]. However, while EV monotherapy leads to >40% response rate in treatment-refractory patients and a meaningful improvement in overall survival, long-term remissions are rare[3–6]. Alternative strategies to target NECTIN4, both in the EV-naive and particularly in the EV-resistant settings, remain largely unexplored and are urgently needed.

Surface protein targeting therapeutics, including chimeric antigen receptor (CAR) T cells, are entering into the oncology arsenal. Multiple CAR T cells are approved for hematologic malignancies and can lead to durable remission rates[7]. CARs are synthetic receptors

---

engineered to recognize target tumor-associated antigens (TAAs) and leverage the potency, expansion and memory of T cells for anti-cancer therapy. They can be engineered against virtually any TAA, with the potential of retaining long-term memory and conferring long-term durability[8,9]. Despite the success of adoptive immunotherapy with CAR T cells in hematologic malignancies, the utility of CAR T cells for solid tumors has been challenging due to multiple factors, including heterogeneous or low expression of TAAs and an immunosuppressive microenvironment leading to poor CAR T cell trafficking, persistence or expansion, among others[10].

One of the fundamental factors influencing the activity of CAR T cells is TAA density[11–14]. Across multiple antigens and malignancies, TAA expression is a key determinant of CAR T cell potency. In contrast to native T cell receptors (TCRs), CARs often fail to exhibit meaningful anti-tumor activity when TAA expression is either below a certain threshold[15], or if TAA expression is too heterogeneous[16–18]. Indeed, low TAA expression on tumor cells represents a mechanism for CAR T cell resistance or escape. Conversely, the requirement for high TAA density may allow for a therapeutic window that helps to limit on-target but off-tumor toxicity in the setting of shared antigens between tumors and normal tissues[12,13,19,20]. Therefore, understanding the mechanisms that cancer cells use to regulate TAA expression is critical not only to improve efficacy but also to understand potential mechanisms of resistance.

We previously showed that *NECTIN4* is enriched in luminal subtypes of UC and that its expression is correlated with multiple luminal transcription factors including *GATA3*, *FOXA1*, and *PPARG* across multiple UC cell lines and clinical cohorts[2]. Interestingly, *PPARG* encodes the nuclear receptor peroxisome proliferator-activated receptor γ (PPARγ), a transcriptional regulator of metabolism whose activity can be modulated via multiple small molecules and by directly binding fatty acids[21]. We therefore hypothesized that modulating this pathway may not only regulate NECTIN4 levels, but also enhance NECTIN4-targeting therapies.

In this study, we develop multiple, second-generation CAR T cell candidates with high specificity against NECTIN4 and establish the potency of our CAR T cells in multiple UC cells expressing a range of endogenous NECTIN4. Focusing on the PPARγ pathway, we identify a mechanism by which UC cells regulate NECTIN4 expression, and we evaluate the effects of PPARγ agonists, which were initially approved for diabetes, on the activity and therapeutic window of NECTIN4-CAR T cells. Finally, we explore the anti-tumor activity of NECTIN4-CAR T cells in an EV-resistant UC model.

## Results

### Development of an antigen-specific NECTIN4-targeting CAR with cytotoxic activity

To explore alternative strategies targeting NECTIN4 that could be leveraged for UC, we engineered and cloned six different NECTIN4 binding scFv sequences into the previously described second generation EQ-28ζ CAR backbone construct containing an IgG4Fc(EQ) spacer, the CD28 transmembrane domain and the CD28 costimulatory domain with CD3ζ signaling domain (Fig. 1a)[22,23]. This backbone was previously used in a CD19-targeted CAR that showed activity against B-cell malignancies[22]. The IgG4-Fc mutant EQ linker exhibits reduced binding to Fcγ receptors (FcγR) and superior anti-tumor efficacy and persistence in immunodeficient mice[24], and co-expresses enhanced green fluorescent protein (eGFP) linked with the T2A self cleaving peptide, which can be used as a transduction marker. CD3+ pan T cells were enriched and transduced with each of the NECTIN4-CAR lentiviral candidates, and anti-tumor activity against the RT112 human UC cell line was evaluated in co-culture assays. We found that the majority (CAR1, CAR2, CAR4, CAR5) but not all of our CAR T candidates had potent anti-tumor activity, and exhibited CD4:CD8 ratios of ~1.5 (Supplementary Fig. 1a, b). We picked the lead candidate based on a

previously published binder[25] (NECTIN4-CAR-1, hence referred to as NECTIN4-CAR) for further characterization.

To evaluate the specificity of the NECTIN4-CAR T cells, we generated NECTIN4 knockout (KO) isogenics in RT112 parental cells using CRISPR/Cas9 and two unique guide sequences (KO1 and KO2). We validated the loss of both total and surface NECTIN4 (Fig. 1b, c), which did not affect cell proliferation (Supplementary Fig. 1c). NECTIN4-CAR T cells exhibited dose-dependent cytotoxicity against RT112 cells over a range of effector to tumor cell (E:T) ratios compared to non-transduced (NTD) T cells from the same donor (Fig. 1d). Importantly, the NECTIN4-CAR T cells did not kill the RT112 NECTIN4-KO cells (Fig. 1e). Killing efficacy was quantified using the kill index, which was calculated by computing the inverse of the area under the curve (AUC) from the growth curves (Fig. 1f). NECTIN4-CAR T cells co-cultured with parental RT112 cells, but not KO cells, demonstrated interferon-gamma (IFNγ) release, an indicator of T cell activation (Fig. 1g). Similar co-culture studies were also performed in UMUC-9 parental and NECTIN4 KO cells, as well as the UMUC-3 cells, which lack endogenous NECTIN4 expression (Supplementary Fig. 1d, e). Consistent with our previous studies, the NECTIN4-CAR T cells exhibited dose-dependent cytotoxicity against the parental but not NECTIN4 KO UMUC-9 cells (Supplementary Fig. 1f, g). Moreover, ectopic overexpression (OE) of NECTIN4 in UMUC-3 cells, which do not express endogenous NECTIN4 in the parental line, was sufficient to activate CAR T killing (Supplementary Fig. 1h, i), and OE of NECTIN4 in UMUC-1 cells also increased CAR T killing (Supplementary Fig. 1j, k). No cytotoxic activity was observed with the non-transduced (NTD) T cells in any co-culture setting. Taken together, these results demonstrate that NECTIN4-targeting CAR T cells have potent and specific activity against NECTIN4-expressing UC cells.

### NECTIN4-CAR T cell potency is dependent on NECTIN4 expression

We previously demonstrated that NECTIN4 expression is heterogeneous across the consensus molecular subtypes of UC and found that NECTIN4 was most enriched in the luminal subtypes[2,26]. We therefore selected a representative panel of human UC cell lines with a range of endogenous NECTIN4 expression. The luminal cell lines (UMUC-1, UMUC-9, RT112), which express PPARγ, displayed varying levels of NECTIN4 while the basal cell lines (UMUC-3 and 647 V) expressed no or low levels of NECTIN4 (Fig. 1h), consistent with our prior study[2].

To assess the antigen density dependence of the NECTIN4-CAR T cells, we performed co-cultures of NECTIN4-CAR T cells with each cell line. We observed a strong positive correlation between killing efficacy of the NECTIN4-CAR T cells and NECTIN4 expression (Fig. 1i, j). This association with expression was also reflected in IFNγ release, an indicator of T cell activation (Supplementary Fig. 1, m). Interestingly, the luminal UMUC-1 cells were more susceptible to CAR T kill compared to the basal 647 V cells, despite having relatively lower NECTIN4 levels, suggesting additional factors in luminally-differentiated cells may mediate sensitivity to CAR T kill, even in low TAA density settings.

### PPARγ agonists and inverse agonists modulate NECTIN4 expression

Given that sensitivity to NECTIN4-CAR T cells correlates with expression of NECTIN4, we sought to identify potential strategies to augment its expression. Based on our previous observation that *NECTIN4* is associated with expression of the luminal transcription factor *PPARG*, we hypothesized that NECTIN4 levels could be modulated by PPARγ agonists and inverse agonists.

To evaluate this, we treated UC cells with rosiglitazone and pioglitazone, two thiazolidinedione (TZD) drugs that are highly selective and potent agonists of PPARγ, which are used for treating diabetes[27]. We found that TZD treatment increased *NECTIN4* mRNA expression (Fig. 2a). Both TZDs also increased PPARγ signaling in UC cells, as evidenced by increased HPGD (15-hydroxyprostaglandin dehydrogenase),

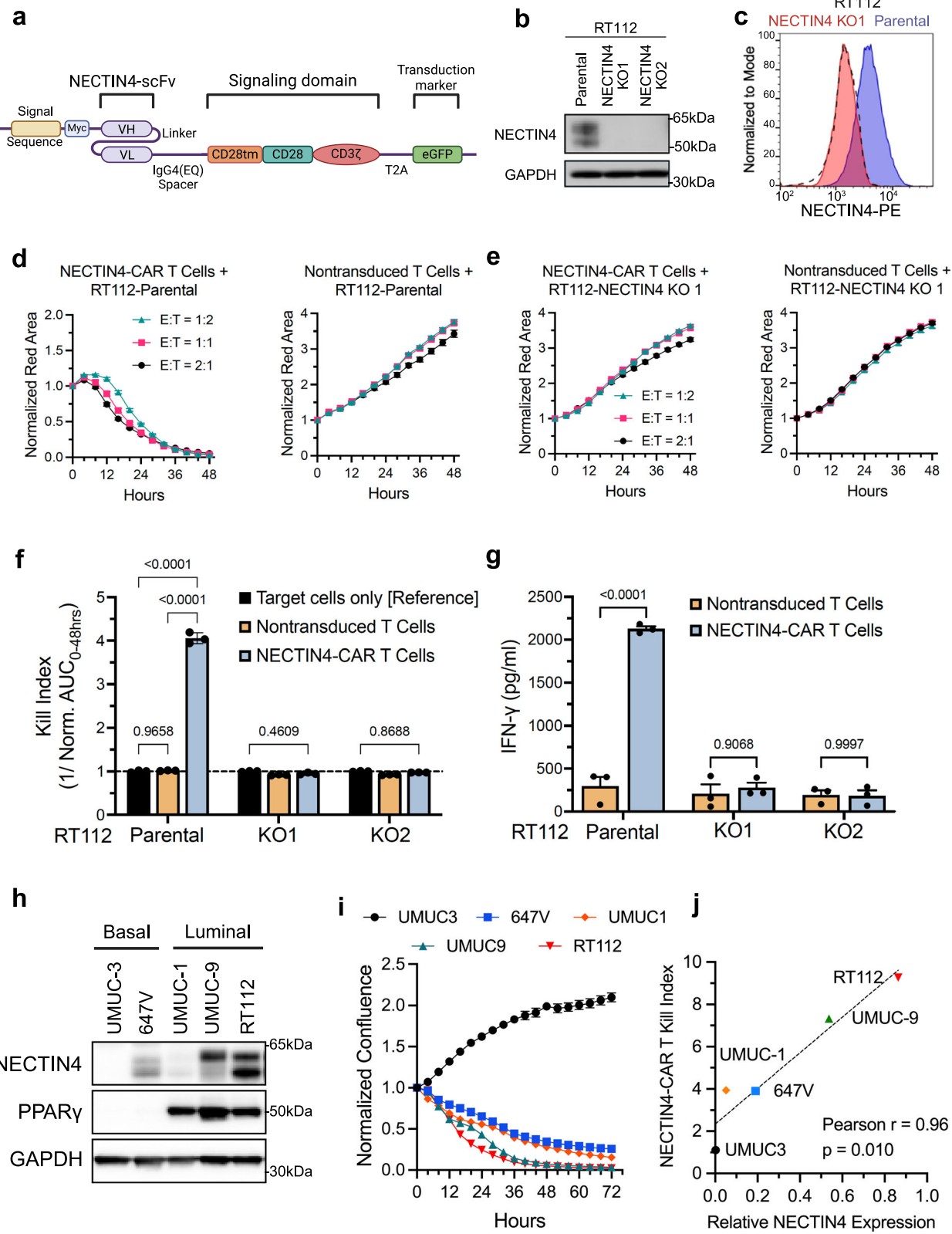

a known downstream transcriptional target of PPARγ and readout of TZD activity (Fig. 2b)[28]. Conversely, T0070907, an inverse agonist of PPARγ, decreased *NECTIN4* mRNA and NECTIN4 protein levels, as well as HPGD (Fig. 2a–d, Supplementary Fig. 2a–e). Rosiglitazone increased surface and total protein expression in a dose-dependent manner across multiple UC cells (Fig. 2c–g, Supplementary Fig. 2a–f), without affecting TROP2, another surface protein target expressed in UC and other

cancers (Fig. 2h)[29,30]. Interestingly, arachidonic acid (AA), a naturally-occurring, polyunsaturated fatty acid that is also a PPARγ ligand[31], was also able to induce NECTIN4 expression (Supplementary Fig. 2g, h). In contrast, treatment with the selective PPARα agonist BMS-687453 did not induce NECTIN4, suggesting some degree of specificity amongst the PPAR family of proteins in controlling NECTIN4 (Supplementary Fig. 2i). Moreover, we used CRISPRi and two unique sgRNAs to genetically

**Fig. 1 | Development of second-generation CD28 chimeric antigen receptors (CARs) targeting NECTIN4 for CAR T cell therapy. a** Schematic of the lentiviral NECTIN4-CAR construct: signal peptide, Myc tag, scFv targeting NECTIN4, CAR T backbone construct containing an IgG4Fc(EQ) spacer, the CD28 transmembrane domain and the CD28 costimulatory domain with CD3ζ intracellular signaling domain, T2A self-cleavage peptide, GFP transduction marker. scFv single-chain variable fragment, VL variable light chain, VH variable heavy chain, tm transmembrane, GFP green fluorescent protein. Created with BioRender. Chang, K. (2025) https://BioRender.com/k3j6fgi. **b** Western blot analysis of NECTIN4 expression in whole cell lysates of RT112 parental cells and NECTIN4 knockout (KO) cells. This was repeated $n = 3$ independent times with similar results. **c** NECTIN4 surface staining in RT112 parental cells and NECTIN4 knockout (KO) cells. **d, e** Growth curves of (**d**) RT112 and (**e**) NECTIN4 KO target cells with fluorescently labeled nuclei (NucLightRed, NLR) co-cultured with NECTIN4-CAR T cells (left) or non-transduced (NTD) T cells (right) at indicated effector-to-target (E:T) cell ratios. **f** A kill index [1/ area under the curve (AUC)] was calculated as a time-dependent measure of killing efficacy against the indicated tumor target cells at an E:T ratio of 1:2. $n = 3$ biological replicates. **g** IFNγ quantification by ELISA from co-cultures of NECTIN4-CAR or NTD T cells with the indicated RT112 cell lines in (**b**) at a 1:2 ratio at 24 h. $n = 3$ biological replicates. **h** Western blots demonstrating NECTIN4 protein expression across luminal and basal human urothelial carcinoma cell lines. This was repeated $n = 3$ independent times with similar results. **i** Growth curves of indicated cell lines co-cultured with NECTIN4-CAR T cells at an E:T ratio of 1:1. A representative experiment of $n = 3$ biologically independent experiments performed in technical triplicates is shown, and error bars represent mean ± SEM. **j** Scatter plot showing the NECTIN4-CAR T kill index against target cell lines in (**h**) versus respective expression of NECTIN4. Pearson's correlation is shown for NECTIN4 expression versus kill index ($r = 0.96$, $P = 0.0097$, two-tailed). For panels (**d–g**) $n = 3$ biological replicates per conditions were used and data are presented as mean ± SEM. For panels (**b,h**), GAPDH was used as a protein loading control. For panels (**f**) and (**g**), two-way ANOVA with Sidak's multiple comparison test was used. Source data are provided as a Source Data file.

inhibit *PPARG* and found that loss of PPARγ decreased NECTIN4 expression as well as FABP4 (fatty acid binding protein 4 or adipocyte protein 2 (aP2)), another downstream target of PPARγ[32] (Fig. 2i). Finally, we identified a predicted PPARγ binding site just upstream of the NECTIN4 transcriptional start site (TSS) and performed chromatin immunoprecipitation (ChIP) followed by PCR, which demonstrated that PPARγ is bound to the NECTIN4 promoter in UC cells (Fig. 2j–l). Taken together, our data demonstrate that in UC cells, the PPARγ pathway regulates NECTIN4 by binding to the NECTIN4 promoter, thereby increasing *NECTIN4* transcription and ultimately leading to more NECTIN4 total and surface protein expression.

## PPARγ agonists prime and enhance sensitivity to NECTIN4-CAR T cells in vitro

Next, we asked whether we could leverage the ability of TZDs to augment NECTIN4 expression in order to enhance NECTIN4-CAR T cell sensitivity, thereby expanding the therapeutic window. To do so, we pre-treated RT112 and two UC lines with low endogenous levels of NECTIN4, UMUC-1, and HT1197, with rosiglitazone for 72 h. After washing out the rosiglitazone, we co-cultured the target tumor cells with NECTIN4-CAR T cells. Across all three UC cell lines, NECTIN4-CAR T cells exhibited enhanced killing efficacy against rosiglitazone pre-treated cells compared to control DMSO pre-treated cells (Fig. 3a–f). The enhanced killing index was seen across a range of E:T ratios (Fig. 3a–f), which also led to higher levels of IFNγ release (Fig. 3g, h). As expected, rosiglitazone pre-treatment of NECTIN4 KO cells did not induce NECTIN4 expression or CAR T killing activity (Supplementary Fig. 2j, 3a–c). We also evaluated NECTIN4-CAR T cells generated from a second donor ("Donor B"), as well as a second CAR construct containing a different NECTIN4 binder (CAR5) and found that the effects of rosiglitazone pre-treatment were robust and generalizable (Supplementary Fig. 3d–i). Finally, given that NECTIN4 is expressed on the normal skin epithelium as well as on normal bladder urothelial cells, we sought to evaluate potential on-target, off-tumor toxicity of the NECTIN4-CAR T cells. To do this, we cultured normal bladder epithelial cells and normal skin keratinocytes, with or without rosiglitazone pre-treatment, and assessed the NECTIN4 levels. We found that bladder epithelial cells expressed lower levels of NECTIN4 than skin keratinocytes, either with or without rosiglitazone pre-treatment, and that both normal cells expressed significantly lower levels than RT112 cancer cells (Supplementary Fig. 3j). While NECTIN4-CAR T cells potently killed RT112 cells, they were inactive against normal skin keratinocytes and normal bladder epithelial cells when co-cultured at similar E:T ratios (Supplementary Fig. 3k). Taken together, these data demonstrate that across multiple UC cell lines, short-term priming with rosiglitazone enhances sensitivity to NECTIN4-CAR T cells, without increasing potential on-target, off-tumor toxicity to normal skin keratinocytes or normal bladder epithelial cells.

## Systemic rosiglitazone increases endogenous NECTIN4 in UC xenograft models

We next tested the ability of TZDs to upregulate NECTIN4 in vivo. We established RT112 xenografts in NOD scid gamma (NSG) mice. When tumors were approximately 50–100 mm³, mice were started on rosiglitazone for 5 days, and tumors were harvested 6–12 h after the last dose (Fig. 4a). Rosiglitazone administration increased NECTIN4 as well as HPGD (which we used as a surrogate marker of PPARγ activity) in RT112 tumor xenografts (Fig. 4b–d). In addition, we detected increased levels of soluble NECTIN4 in the blood (Supplementary Fig. 4a), as well as increased membranous NECTIN4 after rosiglitazone treatment (Supplementary Fig. 4b, c). We also found that rosiglitazone increased NECTIN4 and HPGD in mice bearing HT1197 (Fig. 4h–j) and UMUC-1 tumor xenografts (Supplementary Fig 4d–f). Collectively, these findings demonstrate that rosiglitazone increases NECTIN4 expression in multiple UC tumor xenografts.

## Combining rosiglitazone with NECTIN4-CAR T cell therapy enhances anti-tumor efficacy

To determine whether augmenting NECTIN4 expression with TZDs enhances the therapeutic efficacy of NECTIN4-CAR T cell therapy, we repeated the same rosiglitazone pre-treatment paradigm (Fig. 4a) and then treated the mice with a single, intravenous (IV) injection of either NECTIN4-CAR or NTD T cells from the same donor. Systemic rosiglitazone plus NECTIN4-CAR T cells led to improved RT112 tumor growth inhibition compared to vehicle plus NECTIN4-CAR T cells (Fig. 4e). Short course treatment with rosiglitazone did not alter RT112 tumor xenograft growth when compared to vehicle plus NTD T cells (Fig. 4e), suggesting that rosiglitazone alone does not affect RT112 tumor xenograft growth, but does enhance NECTIN4-CAR T cell efficacy and overall survival (Fig. 4e, f). Tumors isolated at the end of the experiment had overall similar levels of *NECTIN4* as well as other luminal/ basal markers (Supplementary Fig. 4g, h). In addition, mice treated with the combination of rosiglitazone plus NECTIN4-CAR T cells did not show any differences in body weight, suggesting that there was no increased toxicity in the combination arm (Fig. 4g). We also evaluated the combination with rosiglitazone in RT112 tumor xenografts using a second T cell donor (Supplementary Fig. 4i–k) as well as in HT1197 tumor xenografts (Fig. 4k, l), which are slower growing and express lower endogenous levels of NECTIN4. Systemic rosiglitazone plus NECTIN4-CAR T cells also improved tumor growth inhibition in HT1197 tumor xenografts (Fig. 4k), without affecting body weight (Fig. 4l). In contrast to the RT112 tumor xenografts, rosiglitazone pre-treatment slightly inhibited HT1197 tumor xenograft growth compared to the vehicle control. Interestingly, the HT1197 line harbors a RXRα hotspot mutation (S427F/Y) and was previously shown to be sensitive to PPARγ-selective inverse agonists[33,34]. Our results suggest that in this UC cell line, anti-tumor effects may also be achieved with PPARγ-selective

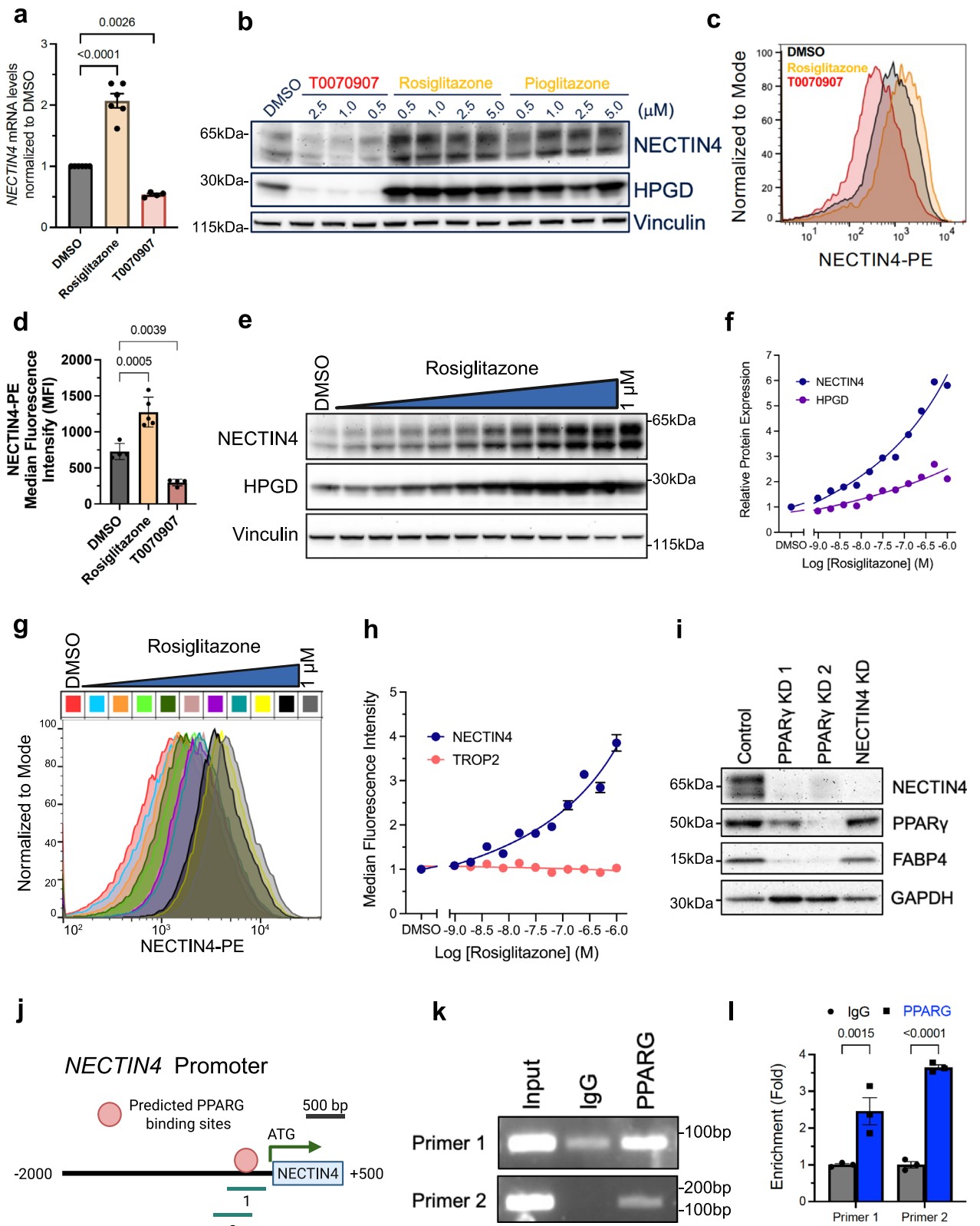

agonists, which may work synergistically with NECTIN4-CAR T cells through surface antigen induction.

## NECTIN4-CAR T cell therapy is effective against a preclinical model of EV resistance

Most patients with metastatic UC who are treated with EV invariably develop resistance to this treatment but the underlying mechanisms of acquired resistance remain to be elucidated. To investigate whether NECTIN4-directed CAR T cell therapy remains effective in the setting of EV resistance, we generated a model of EV resistance in RT112 cells, which underwent repeated exposure to EV at escalating concentrations over the course of 9 months (Fig. 5a). A polyclonal EV resistant (EV Res) cell population was obtained that exhibited a 10-fold higher $IC_{50}$ for EV compared to the parental cell line (Fig. 5b). RT112 EV Res

**Fig. 2 | PPARγ-mediated modulation of NECTIN4 expression. a** *NECTIN4* mRNA levels in RT112 cells treated with 1μM rosiglitazone and T0070907 (T007) for 72 h. Data are presented as mean ± SEM, *n* = 6 for DMSO and rosiglitazone, *n* = 4 for T0070907, all biological replicates. **b** Western blot for NECTIN4 and HPGD in RT112 cells treated with T0070907, rosiglitazone, and pioglitazone at indicated concentrations for 72 h. This was repeated *n* = 3 independent times with similar results. **c, d** NECTIN4 surface staining (**c**) and quantification of median fluorescence intensify (MFI) (**d**) in RT112 cells treated with 1μM rosiglitazone and T0070907 for 72 h. Data are presented as mean ± SEM, *n* = 4 for DMSO and T0070907, *n* = 5 for rosiglitazone, all biological replicates. **e** Western blot for NECTIN4 and HPGD in RT112 cells treated with rosiglitazone for 72 h across a dose series (starting at 1 μM on the right most lane with serial 2-fold dilutions to the left). This was repeated *n* = 3 independent times with similar results. **f** Dose response curves of total NECTIN4 and HPGD protein expression after 72 h rosiglitazone treatment in RT112 cells. Data are presented as mean values. **g** NECTIN4 surface staining in RT112 cells treated with rosiglitazone for 72 h across a dose series. Dose series starts at 1μM with serial 2-fold dilutions. **h** Dose response curves of surface NECTIN4 and TROP2 expression

for rosiglitazone in RT112 cells after 72 h. Data are presented as mean ± SEM, *n* = 2 biological replicates. **i** Western blots for NECTIN4, PPARγ and FABP4 in RT112 cells expressing sgRNAs against *GAL4* (control), *NECTIN4*, and two unique *PPARG* guides. This was repeated *n* = 3 independent times with similar results. **j** Schematic of select predicted PPARG binding sites (labeled 1 and 2) on the NECTIN4 promoter region (500 to −2k) by FIMO using transcription binding motif matrices JASPAR 2018. Created with BioRender. Chang (2025) https://BioRender.com/k3j6fgi. **k** Chromatin from RT112 cells was precipitated using antibodies against PPARG or IgG. Primers targeting putative PPARG binding motifs 1 and 2 from (**j**) were used for PCR and the PCR product was visualized by gel electrophoresis. **l** Quantitative analysis of ChIP-qPCR experiments. Results are represented as fold-enrichment relative to IgG control. Data are presented as mean ± SEM, *n* = 3 biological replicates. For panels (**a**) and (**d**), ordinary one-way ANOVA with Sidak's multiple comparison was used. For panel (**l**), two-sided, unpaired Student's *t* test was used for each primer set. Vinculin shown as a loading control for panels (**b**) and (**e**) and GAPDH for panel (**i**). Source data are provided as a Source Data file.

cells had similar total and surface NECTIN4 expression (Fig. 5c–e), indicating that antigen escape was not the mechanism of resistance in this model. Interestingly, in a cohort of UC patients previously treated with EV who underwent pre-EV and post-EV biopsies, NECTIN4 expression was overall higher at the time of EV resistance and was increased in the majority of matched biopsies evaluated (9 of 17) (Fig. 5f–h). These data suggest that an alternate targeting modality against the same antigen (such as a NECTIN4-CAR T) may have anti-tumor efficacy, even in the setting of EV resistance.

To evaluate this hypothesis, we co-cultured RT112 parental and EV Res cells with NECTIN4-CAR T cells. While the EV Res cells were resistant to EV (Supplementary Fig. 5c, g), they were equally susceptible to the NECTIN4-CAR T cell, which we generated using a different single chain variable fragment (scFv) binder from enfortumab (Fig. 5i, j, Supplementary Fig. 5d, 5h). To account for differences in the binders, we also tested an enfortumab-derived NECTIN4-CAR T cell (NECTIN4-CAR-2) (Supplementary Fig. 5a, e, i). The killing efficacy of NECTIN4-CAR2 T cells was similarly dependent on surface NECTIN4 levels and could be enhanced by pre-treating cells with rosiglitazone (Supplementary Fig. 5b). Importantly, NECTIN4-CAR2 T cells exhibited comparable killing efficacy against RT112 parental and EV Res cells across multiple E:T ratios, except at the highest E:T ratio (Fig. 5k, Supplementary Fig. 5e–i), suggesting that EV Res cells did not lose the enfortumab recognition domain on NECTIN4 as the mechanism of resistance.

We next sought to validate that the RT112 EV Res model also exhibited resistance in vivo. Whereas a single dose of EV led to significant tumor growth inhibition in the RT112 parental cells, the EV Res cells displayed no delay in tumor growth (Fig. 5l), even at a higher EV dose (Supplementary Fig. 5k). Next, we tested the efficacy of systemic NECTIN4-CAR T cell therapy against EV resistant tumors and found that NECTIN4-CAR T cells retained highly potent, anti-tumor activity (Fig. 5m). Finally, we generated a second EV resistant model in the UMUC-1 cell model (UMUC-1 EV RES1, Supplementary Fig. 6a), and found that while UMUC-1 EV RES1 cells had slightly lower levels of NECTIN4 (Supplementary Fig. 6b, c), they remained suspectible to NECTIN4-CAR T cells (Supplementary Fig. 6d, e). Taken together, these results demonstrate that in EV resistant cell line models and a subset of patients who acquire EV resistance but retain tumor NECTIN4 expression, that alternative NECTIN4-targeting strategies such as CAR T cell therapy remain effective, informing future sequencing strategies of these therapies.

## Discussion

Despite considerable progress over the last decade, new treatment options for advanced UC remain a critical unmet need. Here, we designed and evaluated the anti-tumor activity of multiple, second-generation CAR constructs targeting NECTIN4. We showed that

NECTIN4-CAR T cells exhibit high specificity and potency across a range of UC models, including EV resistant cells, and that NECTIN4-CAR T antitumor activity can be enhanced by short-term priming of tumor cells with TZDs such as rosiglitazone. Importantly, our study provides strong rationale for clinical translation to evaluate NECTIN4-CAR T therapy in the frontline and EV-resistant settings, and guides future combination strategies with other NECTIN4-targeting agents.

While there has been considerable progress of using CAR T therapy in hematologic malignancies[35,36], notable challenges remain for solid tumors including target diversity, tumor heterogeneity, tumor penetration and homing, and a hostile tumor microenvironment (TME)[17,18,37]. Current strategies to overcome these challenges have primarily focused on modifications of CAR T cells to enhance function, proliferation or overcome exhaustion[38–42]. With regard to TAA heterogeneity, multi-specific CAR T therapy featuring two or more different CARs in the same T cell such as tandem CARs (TanCARs)[43,44] as well as multivalent CAR T cells are being actively explored (e.g., clinical trials NCT06186401 and NCT04430595). Increasing the affinity of the CAR itself is another strategy, but carries a greater risk of on-target off-tumor toxicity in the setting of low TAA expression in normal tissues[45]. In contrast to increasingly complex CAR designs and T cell manipulation[46], we propose that modulating the tumor cell by transiently upregulating TAA expression is another approach that addresses the problem of expression heterogeneity, particularly among low expressors.

We previously described the heterogeneity of NECTIN4 expression in UC, which is enriched in luminal subtypes and decreased in basal subtypes[2]. In this study, we show that NECTIN4-CAR T cells exhibit cytolytic activity against multiple UC models with a range of endogenous NECTIN4, including low but not negative expressors. Targeting low NECTIN4-expressing tumor cells is particularly relevant for UC, given a recent study showing that membranous surface expression of NECTIN4 decreases with progression to metastatic disease and that low NECTIN4 expression in metastatic biopsies is associated with impaired responses and outcomes with EV treatment[47]. In this study, we also show that upregulating NECTIN4 using rosiglitazone, a PPARγ agonist originally developed for the treatment of diabetes, enhances the efficacy of the NECTIN4-CAR T cell therapy, which is particularly important for tumors with low or moderate NECTIN4 expression, both in the EV-naïve and EV-refractory settings to improve the durability of response. This is consistent with previous reports showing that CAR T cell efficacy is related to the level of TAA expression on tumor cells[13,48]. Our strategy of repurposing rosiglitazone to prime NECTIN4 expression across a population of tumor cells with low to moderate expression significantly enhanced anti-tumor activity of NECTIN4-targeting CAR T therapy. Importantly, our strategy utilizes a transient pulse of rosiglitazone prior to CAR T administration, limiting the potential toxicities of long-term

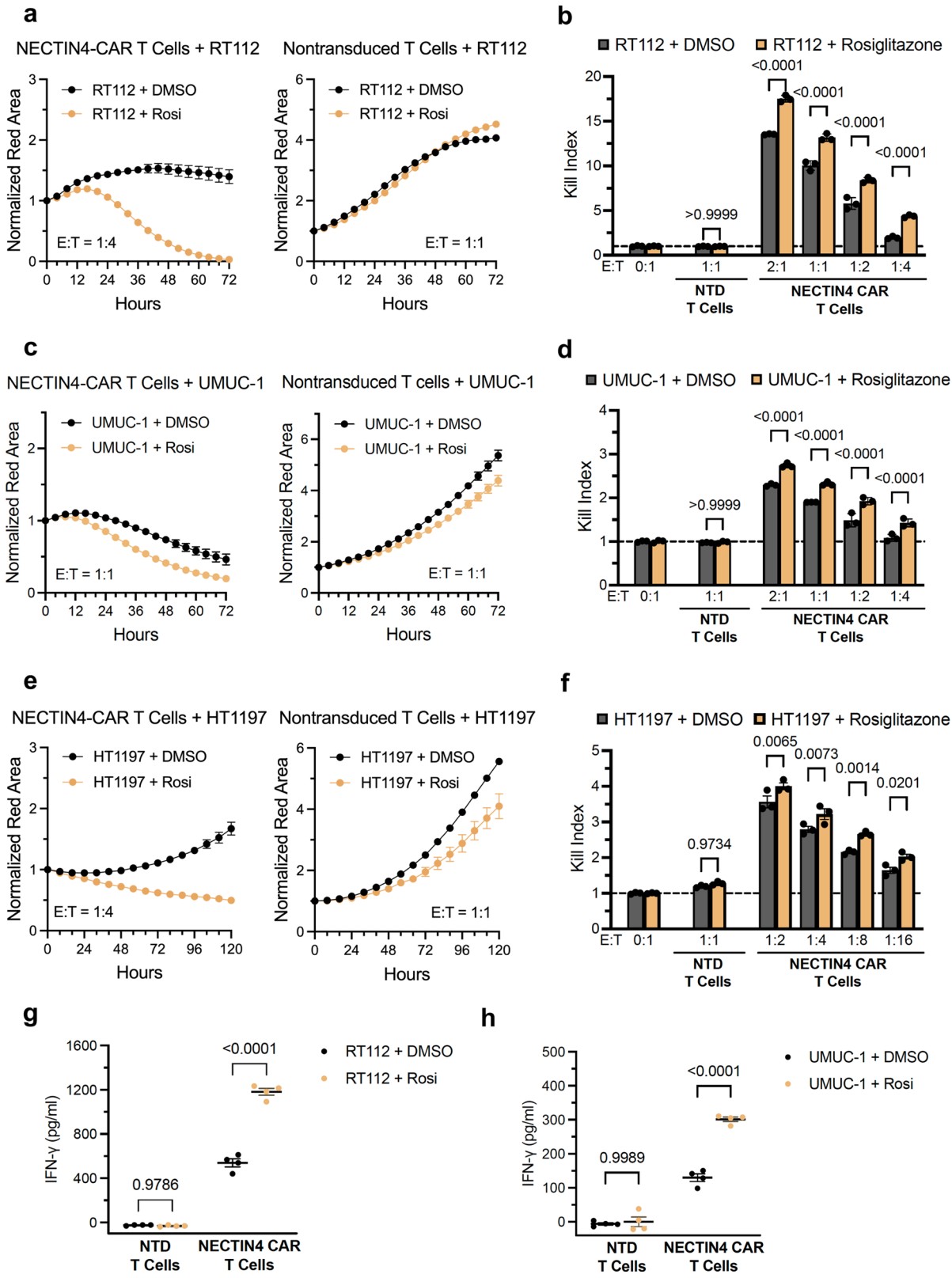

PPARγ agonist use (i.e., edema and cardiotoxicity). Future studies on how rosiglitazone may affect the tumor microenvironment and CAR T cell function should also be pursued.

Our observations are also consistent with previous studies on the critical role of TAA density on other solid and hematologic tumor types, such as ALK in neuroblastoma, GD2 in diffuse midline gliomas or CD72 in B-cell acute lymphoblastic leukemia (B-ALL)[13,14,49]. Indeed,

strategies to upregulate surface ALK using ALK inhibitors in neuroblastoma or to upregulate surface CD72 using SHIP1 inhibitors in B-ALL significantly improves CAR T-mediated killing[48,50]. These studies suggest that surface TAA expression will likely be an important predictive biomarker for CAR T therapies as they enter clinical trials, including those that target NECTIN4, and that strategies to upregulate TAA expression may expand the therapeutic window for these therapies.

**Fig. 3 | Rosiglitazone pretreatment enhances sensitivity to NECTIN4-CAR T cells in multiple UC cell lines.** Representative growth curves of (**a**) RT112, (**c**) UMUC-1, and (**e**) HT1197 cells pretreated for 72 h with DMSO vehicle (black) or 1uM rosiglitazone (gold) and co-cultured with (left) NECTIN4-CAR T cells or (right) NTD T cells at indicated E:T cell ratios. Kill indices for NECTIN4-CAR and NTD T cells against (**b**) RT112, (**d**) UMUC-1, and (**f**) HT1197 cells pretreated for 72 h with DMSO vehicle (black) or 1uM rosiglitazone (gold) across various E:T ratios are shown. For panels **a-f**, data are presented as mean ± SEM, $n = 3$ biological replicates for all groups in (**b**, **d**, **f**). For panels (**b**, **d**, **f**), two-way ANOVA with Sidak's multiple comparison was used. **g,h** FN-γ quantification by ELISA from co-cultures of NECTIN4-CAR or NTD T cells with RT112 (**g**) or UMUC-1 (**h**) cells pretreated for 72 h with DMSO control (black) or 1 μM rosiglitazone (gold) at an E:T ratio of 1:1 at 24 h. Data are presented as mean ± SEM, $n = 3$ biological replicates per condition. For panels (**g**, **h**), two-way ANOVA with Sidak's multiple comparison was used, and data are presented as mean ± SEM, $n = 4$ biological replicates for all groups. Source data are provided as a Source Data file. Created with BioRender. Chang (2025) https://BioRender.com/k3j6fgi.

PPARγ is a transcription factor that functions as a ligand-activated nuclear receptor with well-established roles in regulating glucose and lipid metabolism, adipocyte differentiation, inflammation, and cellular growth and differentiation[21,51]. We found that in UC cells, PPARγ regulates *NECTIN4*, as well as other previously described downstream genes such as *FABP4*[52]. The role of PPARγ signaling pathway in modulating urothelial carcinogenesis is controversial, with studies implicating both tumor suppressive and oncogenic roles for PPARγ[53,54]. For example, amplifications and activating mutations in *PPARG* as well as mutations in its binding partner RXRα, can be found in luminal subtypes of UC[34,54]. During urothelial cell development, *PPARG* drives luminal cell identity, while its expression is downregulated in basal subtypes[55,56]. Interestingly, PPARγ activity may also affect anti-tumor immune surveillance and responses[56,57]. Although thiazolidinediones such as pioglitazone have previously been reported to be associated with an increased risk of UC among diabetic patients who used the drug for >2 years[58,59], this association has been disproven in large population-based studies[60], and no association of increased bladder cancer risk has been shown with rosiglitazone. This suggests that the combination of short-course treatment with PPARγ agonists and NECTIN4-targeting anti-cancer therapy may be safe and feasible in the clinical setting, and warrants further investigation.

As EV assumes a significant role in the frontline treatment setting for advanced UC, resistance to this ADC will inevitably develop earlier for many patients. Given the multi-step mechanism of action of ADCs, there are multiple factors mediating resistance, including impaired drug trafficking, disrupted lysosomal function, and payload-related resistance[61]. In our UC models of acquired EV resistance and retained NECTIN4 expression, we show that CAR T cells remain effective, suggesting non-overlapping mechanisms of resistance, and may therefore be a viable option for EV-resistant tumors. In addition, a recent study demonstrated that a subgroup of UC tumors (up to 26%) harbor NECTIN4 genomic amplifications, which predict response to EV[62]. Tumors with NECTIN4 amplifications may also be exceptional responders to alternative NECTIN4-targeting therapies such as CAR T cells, particularly those that develop resistance to the MMAE payload. Future studies using models with antigen escape will help determine whether treatment with rosiglitazone can rescue expression of NECTIN4 and sensitivity to NECTIN4-targeting therapies.

Notably, the majority of patients diagnosed with bladder cancer have non-muscle invasive disease (NMIBC). For patients with high-risk NMIBC, treatment with bacillus Calmette-Guerin (BCG) is the standard of care but fails in up to 40% of patients[63]. Interestingly, BCG-exposed tumors express high levels of NECTIN4[64]. While radical cystectomy is standard for BCG unresponsive patients, there is significant interest in bladder preservation strategies. Given the recognized immunogenicity of UC, intravesical administration CAR T cell therapy may be a viable strategy, and potentially synergistic with BCG to initiate widespread T-cell infiltration into the tumor microenvironment[65–67]. Intravesical delivery of CAR T cells may also mitigate toxicities associated with systemic CAR T cell therapy and should be further evaluated.

Limitations of our current study include the use of immunocompromised mice (which lack some components of the tumor microenvironment (TME) including myeloid cells, which play a role in regulating immunotherapy responses[68,69]) to evaluate the anti-tumor activity of the CAR T therapy and evaluating tumor responses in non-orthotopic tumor sites. Future studies include developing and evaluating a murinized CAR T system to which we can modulate the tumor immune microenvironment, and evaluating intravesical administration of the CAR T cells. Combinations of NECTIN4-targeting CAR T cells with immunomodulatory agents should also be explored. While the combination of EV and pembrolizumab (an anti-PD1 agent) has been approved in UC[70], a recent study examining the combination of pembrolizumab with EGFRvIII-targeting CAR T cells did not show improved efficacy in glioblastoma[71]. Given that a subset of UC expresses PDL1 and PDL2[72,73], additional work to explore the combination of NECTIN4-CAR T therapy plus anti-PD1 agents is warranted.

In summary, we found that modulating the PPARγ pathway increases NECTIN4 expression, which we leveraged by repurposing a diabetes drug, rosiglitazone, to increase targeting and anti-tumor efficacy of NECTIN4-CAR T cells. We demonstrate significant NECTIN4-CAR T anti-tumor activity in both EV-naïve and EV-resistant settings, and identify a strategy to turn low-expressing tumors into higher-expressing tumors, thereby making them become more susceptible to NECTIN4-CAR T therapy. These preclinical results lay the groundwork for further CAR T cell development in UC, and suggest rational drug combinations that expand the therapeutic window of NECTIN4-targeting therapies.

## Methods

### Animal studies

All animal studies were performed under an approved Institutional Animal and Use Committee (IACUC) protocols AN194778 and AN202104. NSG (NOD/SCID/gamma) mice were housed with *ad libitum* food and water on a 12-hour light cycle at the UCSF Preclinical Therapeutics Core vivarium within the UCSF barrier facility under pathogen-free, ambient temperature and humidity housing conditions. NSG mice were obtained through the UCSF in-house breeding core. For subcutaneous xenografts, $1 \times 106$ cells were injected into the left flank of 8-10 week old male or female NSG mice, which were matched based on the sex of the cell line. The injected cells were resuspended in 1:1 serum-free media and Matrigel (BD Biosciences). Mice were enrolled into treatment groups once tumor volumes reached between 50–100 mm³, typically 7–10 days after tumor cell inoculation. Mice were given rosiglitazone at a dose of 20 mg/kg body weight using oral needle gavage once daily for 5 days or an equal volume of vehicle solvent. $(3–5) \times 10^6$ nontransduced (NTD) control or NECTIN4 CAR T cells were injected intravenously through the tail vein. Tumors were measured with digital calipers and mice were weighed twice weekly by personnel from the UCSF Preclinical Therapeutics Core in a blinded fashion. Tumor volumes were recorded using Studylog Animal Study Workflow software and plotted using Prism (GraphPad, v10). Mice were euthanized when tumors reached 20 mm in any direction or 2000 mm³ volume, whichever occurred first. For survival analysis, a log-rank test was used to compare the overall survival of mice in each cohort.

### Cell culture

All research conducted during this study complies with all relevant institutional ethical regulations and all safety regulations under a Biological Use Authorization (BUA). UMUC-3 cells were a gift from

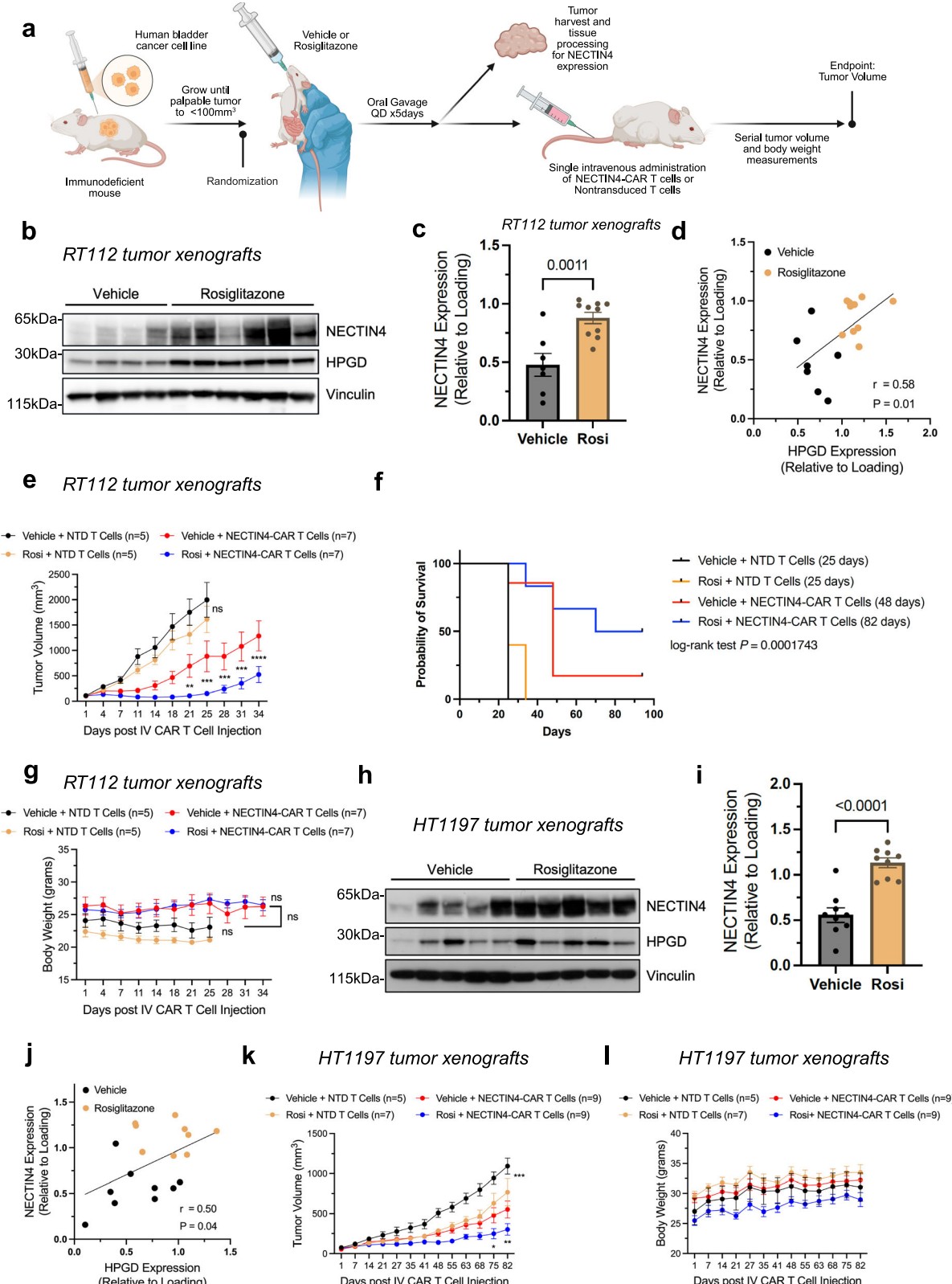

Bradley Stohr (UCSF). RT112, UMUC-1 and UMUC-9 cells were gifts from Drs. Peter Black (University of British Columbia) and David McConkey (Pathology Core, Bladder Cancer SPORE, MD Anderson Cancer Center). HT-1376, HT-1197, and 647 V cells were obtained from the UCSF Cell Culture Facility. Cells were grown in standard MEM media (Corning) supplemented with 10% FBS (Seradigm) and 1% penicillin/streptomycin. Primary human bladder epithelial cells (A/T/

N) and primary human keratinocytes were purchased from the American Tissue Culture Collection (ATCC) or a gift from Dr. Bahram Razani (UCSF), and cultured in either bladder epithelial cell basal medium supplemented with bladder epithelial growth kit (ATCC) or dermal cell basal medium supplemented with keratinocyte growth kit (ATCC), respectively. All experiments were conducted within 30 passages from the parental stock. Cancer cells labeled with NucLightRed

**Fig. 4 | Systemic rosiglitazone treatment primes tumor NECTIN4 expression and enhances anti-tumor activity of NECTIN4-CAR T cell therapy in vivo.**
**a** Schematic of in vivo studies investigating the combination of rosiglitazone plus NECTIN4-CAR T cell therapy. Created with BioRender. Chang (2025) https://BioRender.com/k3j6fgi. Briefly, xenograft models were established by subcutaneous injection of indicated UC cells into NOD/scid/gamma (NSG) mice. When tumors reached approximately 50–100mm³, mice were randomized to receive daily rosiglitazone (20 mg/kg) or vehicle for 5 days, followed by a single injection of 2.5–5 × 10⁶ NECTIN4-CAR or NTD T cells via IV injection. Some tumors were collected immediately following rosiglitazone or vehicle administration for analysis. **b**, **c** Representative western blot (**b**) and quantification (**c**) for NECTIN4 and HPGD in RT112 tumor xenografts from mice treated with either rosiglitazone (gold) or vehicle (black) for 5 days. Data are presented as mean ± SEM, $n = 7$ vehicle and $n = 10$ rosiglitazone, all biological replicates. Vinculin shown as a loading control. This was repeated $n = 3$ independent times with similar results. **d** Scatterplot showing NECTIN4 versus HPGD expression from western blot analysis of tumors in (**b**). Pearson's correlation is shown for NECTIN4 expression versus HPGD expression ($r = 0.58$, $P = 0.01$, two-tailed). Data from $n = 2$ independent cohorts of $n = 4–5$ mice each. **e–f** Tumor growth (**e**) and Kaplan–Meier survival curves (**f**) of RT112 subcutaneous xenografts. Mice were treated for 5 days with either vehicle or rosiglitazone followed by an IV injection of NECTIN4-CAR or NTD T cells. $n = 5$ mice in each NTD T cell group and $n = 7$ mice in each NECTIN4-CAR T cell group. For

panel (**e**), ** indicates $p = 0.0058$ at day 21, *** indicates $p = 0.002$, $p = 0.008$, $p = 0.001$ at days 25, 28, 31, respectively, and **** indicates $p < 0.0001$ at day 34, for Vehicle + NECTIN4-CAR T (red) vs Rosi + NECTIN4-CAR T (blue). Error bars represent SEM. ns not significant, NTD non-transduced. For panel (**f**), the log-rank test was used. **g** Body weights of each group of mice in (**e**) over time. Data are presented as mean ± SEM. **h**, **i** Representative western blot for NECTIN4 and HPGD in HT1197 tumor xenografts from mice treated with either rosiglitazone or vehicle for 5 days. Quantification of NECTIN4 is shown in (**i**). Data are presented as mean ± SEM, $n = 9$ biological replicates in each group. (**j**) Scatterplot showing NECTIN4 versus HPGD expression from western blot analysis. Pearson's correlation is shown for NECTIN4 expression versus HPGD expression ($r = 0.50$, $P = 0.04$, two-tailed). Data from $n = 2$ biologically independent cohorts of $n = 4–5$ mice each. **k**, **l** Tumor growth curves (**k**) and body weights of each group (**l**) of HT1197 tumor xenografts treated with vehicle or rosiglitazone followed by single IV injection of NECTIN4-CAR T cells or NTD T cells. The number of mice in each group is shown in parentheses: $n = 5$ mice (Vehicle+NTD), $n = 7$ mice (Rosi+NTD), $n = 9$ mice (Vehicle+NECTIN4-CAR T and Rosi+NECTIN4-CAR T). *** indicates $p = 0.004$ at day 82 for Vehicle + NTD vs Rosi + NTD and ** indicates $p = 0.0068$ at day 82 and * indicates $p = 0.0167$ at day 75 for Vehicle+NECTIN4-CAR T (red) vs Rosi+NECTIN4-CAR T (blue). Data are presented as mean ± SEM. For panels (**c,i**), two-sided, unpaired Student's $t$ test was used. For panels (**e**, **g**, **k**, **l**), mixed effects model with Sidak's multiple comparison test was used. Source data are provided as a Source Data file.

(Sartorius) were transduced according to the manufacturer's instructions, and isolated using puromycin (2μg/ml) or by FACS (BD Fusion). Bladder cancer cells were validated by STR profiling and routinely tested for mycoplasma (Lonza).

## Cloning of CAR constructs
NECTIN4-targeting scFv sequences derived from human anti–nectin-4 monoclonal antibodies N41[25] and AGS-22/Enfortumab[74] were cloned into the BamHI site of a 28z CAR backbone containing IgG4 spacers, CD28 transmembrane domain, CD28 costimulatory domain, CD3ζ chain, EGFP, and WPRE. The heavy (VH) and light (VL) chains sequences were cloned in two configurations (VH-VL and VL-VH) into the CAR backbone plasmid using Gibson Assembly protocol according to the manufacturer's instructions and lentivirus generated as below.

## Lentivirus production
HEK293T-Lenti-X cells (Takara Bio) were thawed, cultured, and expanded in DMEM media supplemented with 10% FBS. HEK293T-Lenti-X cells were transfected with either the NECTIN4 CAR or the NECTIN4-V5 overexpression lentiviral plasmid and the packaging plasmids psPAX2 and pVSVG using TransIT-LT1 transfection reagent (Mirus Bio) or calcium phosphate buffer. Cell supernatant was collected at 48 h and 72 h. The virus was filtered and concentrated using the Lenti-X Concentrator (Takara Bio) according to the manufacturer's instructions and resuspended in serum-free media.

## Human NECTIN4-CAR T cell production
Human T cells were isolated from a healthy human donor peripheral blood Leukopak (Stemcell Technologies, Cat # 70500.2), processed using the EasySep Human CD3 + T Cell Isolation Kit and grown in either ImmunoCult-XF T Cell Expansion Medium (Stemcell Technologies, 01981) or TexMACS™ Medium (Miltenyi Biotech, 130-097-196). Base media was supplemented with human recombinant IL-15 (Stemcell Technologies, 78031) and IL-7 (Stemcell Technologies, 78053) at 10 ng/ml. T cells were then plated on retronectin coated plates (Takara, T100A), stimulated with Human CD3/CD28 T Cell Activator (Stemcell Technologies, 10971) per million cells, and concentrated lentivirus was added. Cells with virus were spun at 1000 rpm for 45 min. After 72 h of incubation, virus was removed and cells were allowed to recover for 2-3 days. Transduction efficiency was evaluated via flow cytometry for GFP. If transduction efficiency was <40% based on GFP positivity, T cells were sorted by FACS (BD Aria) or MACs using a biotinylated c-myc antibody (Miltenyi Biotec, 130-124-877) and isolated using the

MiniMACS separator and columns (Miltenyi) according to the manufacturer's protocol. CAR T cells were counted every 2-3 days and maintained at a density of 10⁶ cells/ml of media. CAR T cells were expanded up until day 20 following activation. The non-transduced (NTD) T cells were not exposed to lentiviral supernatant and did not undergo spinoculation. Otherwise, the NTD T cells were stimulated and expanded in the same manner as the CAR T cells. Activated and expanded T cells were frozen using Cytiva Hyclone 2X (Fisher) and stored at −80 °C. T cells were thawed into media supplemented with IL-15 and IL-7 per above and recovered for 1 day before in vitro or in vivo anti-tumor experiments. Immunophenotyping of CAR T cells was verified by flow cytometry using fluorescently-labeled antibodies to CD3, CD4, and CD8 (all from BD Biosciences).

## IncuCyte CAR T co-culture assays
CAR T cell functional killing assays were performed by co-culturing target bladder cancer cells labeled with NucLightRed (NLR) with human non-transduced (NTD) or NECTIN4-CAR modified T cells at variable effector-to-target (E:T) ratios in 96-well plates. On day 0, 2000-5000 target cells were plated and allowed to adhere overnight. On day 1, effector T cells were added into wells with tumor cells at specified E:T ratios after obtaining a point scan to determine an accurate NLR+ target cell count. Cells were monitored using IncuCyte S3 (Sartorius) and images were obtained every 3-6 h over 72-120 h. Viable target cells were quantified based on the red object count or red area confluence and normalized to the starting day 1 values. The area under the curve (AUC) was calculated to capture the cytolytic activity of CAR T cells against target cells in a time-dependent manner. Kill efficacy was summarized by calculating the kill index, which is calculated as 1/AUC as shown previously[75]. The AUC for a target cell condition was normalized to its internal control of corresponding tumor cells alone without any effector cells in co-culture (E:T = 0:1) to account for any differences in growth rates. Values were plotted using Prism (GraphPad, v10).

## NECTIN4 knockout and overexpression cells
For generating RT112 and UMUC-9 NECTIN4 KOs, parental RT112 or UMUC-9 cells transfected (Lipofectamine 3000) with PX458 (a gift from Dr. Feng Zhang, Addgene #48138) containing one of the following sgRNA targeting sequences: (1) 5'-CATGTGAGCCCGGCTTACGA; (2) 5'-CCAGCTCACCCGCGGGGCAC or a scramble sgRNA (as control). 48–72 h after transfection, GFP-positive cells were sorted by FACS (BD Fusion) and expanded. Cells were then stained with a NECTIN4

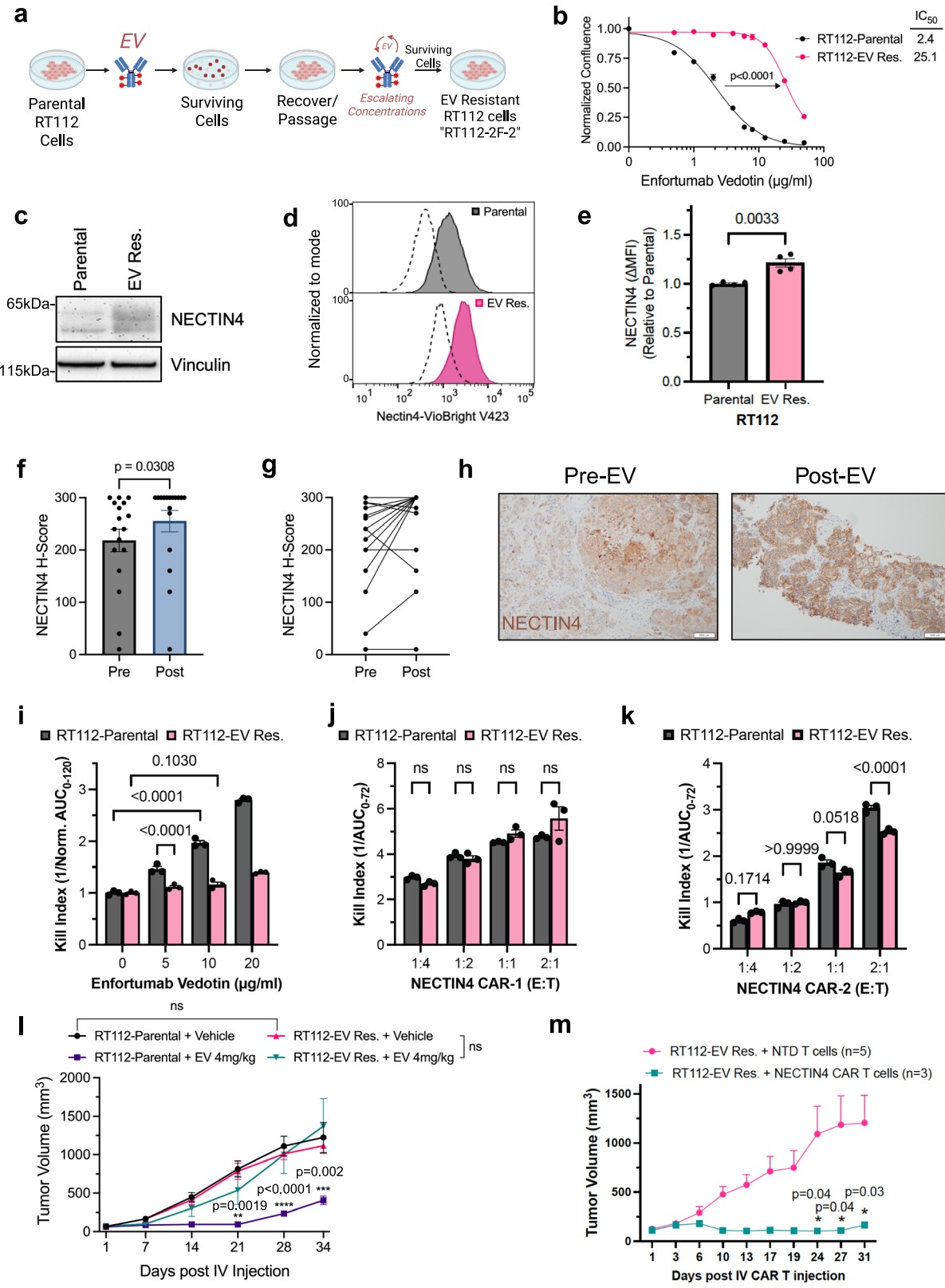

antibody (Miltenyi, clone REA967, 1:100), sorted a second time by FACS (BD Fusion), and negative cells were collected and expanded. For generating NECTIN4 overexpressing cells, the NECTIN4 open reading frame (Horizon, clone ID 100006134) was cloned into lentiviral vectors pCMV-V5-Blast (a gift from Dr. Minkyu Kim) using Gateway cloning protocols (ThermoFisher). Lentivirus was generated as above and cells were transduced with concentrated virus for 3 days, and then selected

in blasticidin (3-5 µg/ml) for at least 5 days. Overexpression was validated by WB and by surface protein staining and flow cytometry using the NECTIN4 antibody (Miltenyi, clone REA967, 1:100).

**Generation of enfortumab vedotin (EV) resistant cells**
RT112 cells were repeatedly exposed to EV (SeaGen) for 3-5 days, with dose escalation of 5 µg/ml per treatment cycle, for a total of 6

**Fig. 5 | NECTIN4-CAR T cell therapy demonstrates efficacy in vitro and in vivo against a preclinical model of enfortumab vedotin (EV) resistance. a** Schematic of parental RT112 cells undergoing cycles of treatment with enfortumab vedotin (EV) at escalating doses to 25 µg/ml, yielding EV resistant cells ("RT112-EV Res."). Created with BioRender. Chang (2025) https://BioRender.com/k3j6fgi. **b** EV dose-response curves for RT112-Parental and RT112-EV Res. following 9 days of treatment with EV, showing EV IC50 for RT112-parental (2.4 µg/ml) and RT112-EV Res. (25.1 µg/ml). **c** Western blot for NECTIN4 in RT112-parental and RT112-EV Res. cell lysates. Vinculin shown as a loading control. This was repeated $n = 3$ independent times with similar results. **d, e** Representative flow cytometry (**d**) and quantification of surface NECTIN4 MFI (**e**) in RT112-parental and RT112-EV Res. cells. Data are presented as mean MFI ± SEM, $n = 4$ biological replicates, and two-sided, unpaired Student's $t$ test was used. **f, g** H-scores for surface NECTIN4 IHC on matched samples taken from a cohort of $n = 17$ UC patients who underwent pre-EV and post-EV biopsies. Data are presented as mean ± SEM in (**f**), and a two-sided, paired Student's t test was used. **h** Example of NECTIN4 stain on a paired biopsy taken pre-EV (H-score=160) and post-EV (H-score=300). Scale bar denotes 1 mm. Kill indices

of (**i**) EV, (**j**) NECTIN4-CAR-1 T cells and (**k**) enfortumab-derived NECTIN4-CAR-2 T cells against RT112-parental (black) and RT112-EV Res. (pink) cells. Error bars represent SEM, $n = 3$ biological replicates for each group. (**l**) Tumor growth curves of RT112-parental and RT112-EVRes. tumor xenografts treated with a single dose of EV (4 mg/kg) or vehicle. RT112-parental ($n = 10$ mice vehicle, black), RT112-EV Res. ($n = 10$ mice vehicle, pink), RT112-parental + EV ($n = 4$ mice, purple), RT112-EV Res. + EV ($n = 5$ mice, teal). Data are presented as mean ± SEM. $p$ values comparig RT112-parental vehicle (black) vs EV (purple) are shown for indicated time points. **m** Tumor growth curves of RT112-EVRes. tumor xenografts in NSG mice treated with a single dose of $5 \times 10^6$ NECTIN4-CAR or NTD T cells. $n = 5$ mice (NTD T cells, pink), $n = 3$ mice (NECTIN4-CAR T cells, teal). Data are presented as mean ± SEM. $p$ values at days 24, 27 and 31 are shown. For panel (**b**), a nonlinear regression model was used to compare the dose-response curves by the extra sum-of-squares F test. For panel (**g**), a paired, two-tailed Student's $t$ test was used. For panels (**i**–**k**), two-way ANOVA with Sidak's multiple comparison test was used. For panels (**l**, **m**), a mixed effects model with Sidak's multiple comparison test was used. Source data are provided as a Source Data file.

treatment cycles. Cells were allowed to recover in between each treatment cycle, prior to the next treatment cycle with EV. After the 6th round of EV at a maximum dose of 25 µg/ml, the EV dose response of RT112-EV-2F-2 generation was assessed.

## Pharmacological compounds and dose response assays
Cells were treated for 48 to 72 h with indicated concentrations of rosiglitazone, pioglitazone, T00700907, or BMS-687453 (all purchased from MedChemExpress). Dimethyl sulfoxide (DMSO) was used as the vehicle. Compounds were added to standard complete MEM medium containing 10% FBS and 1% penicillin/streptomycin. Drug-containing media was aspirated and cells were washed with D-PBS prior to profiling or seeding for functional assays. For EV dose response assays, cells were seeded in a 96-well plate in triplicate. After 24 h, EV was added at indicated concentrations for 7-12 days, and cell number was monitored using an IncuCyte S3 (Sartorius).

## CRISPR Interference Knockdown
CRISPRi cell lines were generated as previously described[2]. Briefly, RT112 cells were transduced with dCas9-KRAB (a gift from Drs. Stanley Qi and Jonathan Weissman, Addgene #46911) and sorted for BFP positivity. The sgRNA guide sequences targeting *PPARG* or *NECTIN4* were ordered from Integrated DNA Technologies and cloned into pCRISPRia_v2 (a gift from Dr. Jonathan Weissman Addgene #84832) using BstX1 and Blp1 and Quick Ligation Kit (New England BioLabs). The sgRNA guide sequences for inhibiting *PPARG* were as follows: (1) 5'-GGCCAACTCACCTAAGGAAA; (2) 5'-GAGTGAGCCAACTCACCTA. The sgRNA guide sequences for *NECTIN4* was as follows: 5'-GACGCTGTGGCCAACAACGA. Lentivirus was made as previously described and cells transduced for 3 days and then selected with 2 µg/ml puromycin for at least 3 days.

## Quantitative PCR
Total RNA was isolated from cells using the Quick RNA Kit (Zymo Research). cDNA was synthesized using the Superscript III RT First Strand Kit (Invitrogen). QPCR was performed using SYBR Green master mix (Roche) in Applied Biosystems QuantStudio6 or QuantStudio7. Ct values were normalized to TBP, Actin, HPRT and GAPDH, and relative expression was calculated using the 2ddCt method. Primer sequences were found using the Harvard Primer Bank (Supplementary Table 1).

## Patient biopsy samples
All patients provided informed consent for research use of tissue prior to biopsy, which were performed at Memorial Sloan-Kettering Cancer Center. Biopsies were collected as standard of care. Additional research slides were collected and stained under MSKCC IRB protocols #12-245 and 06-107.

## Immunostaining and histology
Tissue samples were fixed in 4% PFA overnight and paraffin embedded. Standard hematoxylin and eosin (H&E) staining was performed for routine histology. Antigen retrieval for immunohistochemistry was performed using the BioSB TintoRetriever and citrate buffer. The ABC-HRP Amplification Kit (VectorLabs) was used according to the manufacturer's specifications. Primary antibodies were incubated overnight and secondary antibodies were incubated for 2 h. The following antibodies were used at the indicated concentrations: Normal Rabbit IgG (#3900, 1:100), anti-NECTIN4 (abcam #192033, 1:100), Biotinylated anti-rabbit (Jackson #111-065-144, 1:1000). The DAB developing kit (VectorLabs) was used according to instructions. NECTIN4 H-scores were determined by two board-certified pathologists.

## Surface protein staining and Western Blot
Adherent cells were trypsinized and washed, and then incubated with anti-NECTIN4 PE antibody or anti-NECTIN4 VioBright V423 (Miltenyi Biotec, clone REA967, lot #1324070142, 1:100) for 30-60 min on ice. Cells were analyzed using an Attune NxT Flow Cytometer and the median fluorescence intensity (MFI) was calculated and data were analyzed using FlowJo software. For western blotting, cells were lysed in RIPA buffer containing Halt protease and phosphatase inhibitor cocktail (Thermo Scientific). Lysates were subjected to SDS-PAGE, transferred to PVDF membranes, blocked in 5% w/v BSA, and incubated with primary antibody overnight. The next day, Horseradish Peroxidase (HRP) conjugated secondary antibodies were used. The blot was visualized using ECL Detection Reagents after 24 h (Genesee Scientific). Antibodies against GAPDH (Cell Signaling Technology, #2118, clone 14C10, 1:4000), Vinculin (Cell Signaling Technology, #13901, clone E1E9V, 1:2000), NECTIN4 (abcam #192033, clone EPR14613-68, 1:1000), PPARγ (Cell Signaling Technology, #2435, clone C26H12, 1:1000), HPGD (R&D Systems, AF5660, 1:1000), FABP4 (Cell Signaling Technology, #2120, 1:1000), HRP-anti-rabbit (Cell Signaling Technology, #7074, 1:8000) and HRP-anti-goat (abcam, #ab6741, 1:8000) were used according to the manufacturer's recommended dilutions.

## IFNγ Quantification and NECTIN4 ELISA
To assess IFNγ release, 96-well plates with co-cultures described above were centrifuged and 25−50 µl of co-culture supernatants were collected at 24 and/or 48 h of co-culture and stored at −80 °C. Samples were thawed and analyzed by sandwich ELISA for IFNγ using the BD Human IFN-gamma ELISA Set (BD Biosciences, 555142) according to the manufacturer's protocol. For NECTIN4 ELISA, mouse serum was collected after control or rosiglitazone treatment, diluted and processed using the human NECTIN4 Quantikine kit (R&D, DNEC40) according to the manufacturer's instructions. Plates were read at 450 nm and 570 nm wavelengths using a Biotek Synergy plate reader.

## Chromatin immunoprecipitation (ChIP)-qPCR

RT112 cells were grown to 90% confluence, trypsinized, washed, and collected. SimpleChIP Enzymatic Chromatin IP Kit (Cell Signaling Technology) was used following the manufacturer's protocol. ChIP was performed with anti-PPARG antibody (Cell Signaling Technology). For a negative ChIP control, we used normal rabbit IgG provided in the kit. Primers were designed based on the predicted PPARG binding motifs for qPCR and PCR amplification. Sequences for PCR/qPCR as indicated in Supplementary Table 2. ChIP/ChIP-qPCR experiments were repeated three times for each target motif.

## Reporting summary

Further information on research design is available in the Nature Portfolio Reporting Summary linked to this article.

## Data availability

The source data file for plots and western blots are provided with this paper. Plasmids, lentiviral constructs and cell lines described in this study are available upon request from the corresponding author and subject to an executed Materials Transfer Agreement with the University of California, San Francisco. The raw human biopsy material and data are not available due to patient privacy issues and compliance with the HIPAA (Health Insurance Portability and Accountability Act). Source data are provided with this paper.

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

## Acknowledgements

This work was supported by the NCI/NIH (K08CA273514 to J.C. and K12CA260225 Career Development Award to C.E.C. (PI: Bergsland)), the Bladder Cancer Advocacy Network (BCAN) Young Investigator Award (to J.C.), funds from the UC Cancer Research Coordinating Committee of the University of California, Grant number C25CR8622 (to J.C.), the UCSF School of Medicine Dean's Fellowship (to K.C.), the Fundación Alfonso Martín Escudero (to A.C.G.) and philanthropic gifts from TS and Sharon Ng and the Speier Family (to T.W.F.). We thank Raunak Shrestha and members of the Feng Lab for bioinformatics assistance and critical

reading of the manuscript. This was supported in part by the University of California, San Francisco Preclinical Therapeutics Core (PTC) and UCSF Helen Diller Family Comprehensive Cancer Center Laboratory for Cell Analysis (LCA) and Flow Cytometry Core under NIH/NCI award P30CA082103. Some figures were created or assembled using BioRender.com (under license Chang, K (2025) https://BioRender.com/k3j6fgi).

## Author contributions

K.C., H.M.D., E.Y., C.K., J.Z., R.L., S-Y. L, S.C.B., A.C-G, V.S., and J.C. acquired, analyzed and interpreted the data. M.B., G.O., M.Y.T., D.B.S., J.E.R., H.A-A., and C.E.C. provided and performed analysis and interpretation of biopsy samples. C.C.K.D., E.C., S.P.P., V.S.K., T.W.F., F.Y.F. J.C. provided funding and interpreted the data. J.K.L and A.P.W. provided critical reagents and interpreteted the data. K.C. and J.C. conceived and designed the study and wrote the manuscript. J.C. supervised the study. All authors contributed to editing of the manuscript.

## Competing interests

D. Solit has consulted/received honoraria from Rain Pharmaceuticals, Pfizer, Fog Pharma, PaigeAI, BridgeBio, Scorpion Therapeutics, FORE Therapeutics, Function Oncology, Pyramid, Elsie Biotechnologies, Inc, and Meliora Therapeutics, Inc, all of which are outside the submitted work. H. Al Ahmadie has consulted for AstraZeneca and Paige.AI, all of which are outside the submitted work. J. Rosenberg served in a consulting or advisory role for Aktis, Astellas, AstraZeneca, Bayer, Boehringer Ingelheim, Bristol-Myers Squibb, Century Therapeutics, EMD Serono, Gilead, Century Therapeutics, Lilly Oncology, Pfizer, Roche/Genentech, Merck, Samsung Bioepis, Tyra Bioscience, and Seattle Genetics; has received research support from Astellas, AstraZeneca, Seattle Genetics, Genentech/Roche, Acrivon, and Lilly Oncology, all of which are outside the submitted work. C.K. Ding has consulted for Intuitive and reports research funding from Bristrol Myers Squibb, all of which are outside the submitted work. S.P. Porten reports research funding from Photocure and Kdx, honoraria from Fergene and CG Oncology, and serves on the steering committee or advisory board for Janssen and Vesica Health, all of which are outside the submitted work, all of which are outside the submitted work. T.W. Friedlander reports research funding from Seagen/Pfizer and Roche/Genetech, and has served in a consulting role for Astellas, Merck, Seagen/Pfizer, Gilead, Bicycle Therapeutics, Atkis Oncology, Bristol Meyers Squibb, and Abbvie, all of which are outside the submitted work. V.S. Koshkin has served in a consulting or advisory role for Astellas, Bicycle Therapeutics, Janssen, EMD Serono, Loxo Oncology, MSD, Seagen/Pfizer; has received institutional research funding from Endocyte/Novartis, Curium, Nektar, Gilead, Taiho, Merck and Seagen/Pfizer; and individual research funding from Eli Lilly, all of which are outside the submitted work. F.Y. Feng reported personal fees from Bluestar Genomics, Astellas, Foundation Medicine, Exact Sciences, Tempus, POINT Biopharma, Janssen, Bayer, Myovant, Roivant, SerImmune, Bristol Meyers Squibb, Novartis, and personal fees from POINT Biopharma and other support from Artera, prior to his death, all of which are outside the submitted work. J.K. Lee holds equity in, is on the scientific advisory board of, and receives research funding from PromiCell Therapeutics, outside the submitted work. He is an inventor on International Patent Application No. PCT/US2023/074229 related to NECTIN4 chimeric antigen receptor T cell therapy. A.P. Wiita reports being an equity holder in Indapta Therapeutics and speaker honoraria from Sanofi and AstraZeneca, all of which are outside the submitted work. J. Chou reports consulting fees from Exai Bio and Bicycle Therapeutics outside the submitted work. K. Chang and J. Chou have filed a provisional patent through the University of California Office of Technology and Management based on this work. All other authors report no disclosures.

## Additional information

[1]Helen Diller Family Comprehensive Cancer Center, University of California, San Francisco, CA, USA. [2]Division of Hematology/Oncology, Department of Medicine, University of California, San Francisco, CA, USA. [3]Department of Urology, University of California, San Francisco, CA, USA. [4]Department of Laboratory Medicine, University of California San Francisco, San Francisco, CA, USA. [5]Department of Radiation Oncology, University of California, San Francisco, CA, USA. [6]Division of Hematology/Oncology, Department of Medicine, University of California Los Angeles, Los Angeles, CA, USA. [7]Department of Pathology and Laboratory Medicine, Memorial Sloan Kettering Cancer Center, New York, NY, USA. [8]Genitourinary Oncology Service, Department of Medicine, Memorial Sloan Kettering Cancer Center, New York, NY, USA. [9]Human Oncology and Pathogenesis Program, Memorial Sloan Kettering Cancer Center, New York, NY, USA. [10]Department of Pathology, University of California, San Francisco, CA, USA. [11]Department of Pathology, Stanford University, Stanford, CA, USA. [12]Deceased: Felix Y. Feng. [13]These authors contributed equally: Henry M. Delavan, Elizabeth Yip. ✉e-mail: jonathan.chou@ucsf.edu

