## [Transparent Peer Review file · Nature Communications]

Modulating the PPAR γ pathway upregulates NECTIN4 and enhances chimeric antigen receptor (CAR) T cell therapy in bladder cancer

Corresponding Author: Dr Jonathan Chou

Version 0:

Reviewer comments:

Reviewer #1

(Remarks to the Author)

This manuscript by Chang et al aims to evaluate a NECTIN-4 CAR T cell in targeting EV-refractory bladder cancer patients. The main strength of this manuscript is the elegant demonstration of a mechanistic connection between NECTIN-4 transcription and PPAR-gamma, a pathway when induced, known to drive urothelial cell differentiation. They propose that Rosiglitazone-induced NECTIN-4 expression synergizes with NECTIN-4 CAR T cells to enhance its killing, with potential value in EV-resistant patient populations.

1. The major innovation is Figure 2, showing PPAR-g agonists nicely induces NECTIN-4 protein expression while genetic knockout of PPAR-g decrease NECTIN-4 expression; however, it is unclear whether such regulation can be generalized to other high-grade tumor models.
2. We commend the authors in generating a specific NECTIN-4 CAR T cells, which is valuable and technically promising. However, Nectin-4 is highly expressed in proximal digestive tract, skin, and bone marrow. It is unclear how these authors plan to circumvent the potential toxicities of NECTIN-4 CAR T cells in human patients beyond this preclinical model?
3. In figure 4e and 4j, it is slightly surprising that the response of RT112 and HT1197 to NECTIN-4 CAR-T alone is comparable when the baseline protein expression of NECTIN-4 seems to be much higher in HT1197 vs RT112. Also, these efficacies are not very encouraging when other TME components, e.g., myeloid are not present. However, it is quite surprising that the response of RT-112-EVRes is quite remarkable when NECTIN-4 protein expression (Figure 5c) is quite marginally increased. Can the authors provide some control experiments or explanation/rationale?
4. In figure 4e and 4j, although rosiglitazone demonstrated added effect when administered together with NECTIN-4 CAR-T. The tumors eventually bounced back. Since NECTIN-4 might be associated with differentiated tumor cells, can be co-stain for stem vs differentiated cell markers in association with NECTIN-4 in the residual tumors?

Reviewer #2

(Remarks to the Author)

Chang et al. generated CAR T cells targeting NECTIN4 surface protein which is also used for a new targeted therapy of bladder cancer (BC) using the antibody-drug-conjugate (ADC) enfortumab-vedotin (EV; Padcev). This is a smart idea for 2 reasons: 1) EV has an all comer approval by FDA and EMA based on the argument that >80% of bladder cancer tissues are NECTIN4 positive. Thus, CAR T cells recognizing NECTIN4 should be an efficient tool in the majority of BC patients (both muscle-invasive and non-muscle invasive BC). 2) CAR T cells based therapy for solid cancer are still a bit "left behind" due to some difficulties. Thus it is important to further investigate and promote such approaches. In addition, the authors demonstrate that PPAR γ agonists can augment expression of the target protein. Concurringly, pre-treatment of NECTIN4 low BC could increase target abundance for CAR T and augment treatment efficacy in such cases. This is a sound study using many different control experiments and state of the art methodology and is of high significance for two research communities: 1) those aiming at development of new immunotherapeutic approaches for BC and 2) those trying to move forward CAR T cell based therapeutic approaches for solid cancers. Published data on CAR T cell approaches for BC and on CAR T cells against NECTIN4 is scarce.

Thus, I recommend minor revision.

My comments are:

- One major point refers to authors' argument that combined treatment with the PPAR γ agonists and CAR T cells targeting NECTIN4 could be beneficial for EV-refractory patients: 1) However, they mention themselves that a post-EV biopsy cohort retained NECTIN4 expression and also that their EV-resistant cell line model retained the target protein. Following this argument such EV-refractory cases would not have strong need to further increase NECTIN4 levels. In addition, in our hands killing efficacy of CAR T cells against BC cell lines in vitro was not so much dependent on target density (as mentioned by the authors) and also cell lines with less prominent expression were sufficiently killed. Thus I don't think that it is "given" that high TAA density is generally needed (introduction and discussion p.8 line 296), please rephrase. Instead I would rather think that addition of a PPAR γ agonist would be beneficial in the about 20% of BC patients with low or no NECTIN4 abundance or those losing the protein during cancer progression as reported by Klümper et al 2023. This idea would extend suitability of NECTIN4 CAR T therapy to almost all BC patients, which is even better. So maybe you want to reconsider your arguments concerning the EV-refractory setting.
- Please also discuss further retained NECTIN4 in the mentioned post EV cohort and reduced expression in the Klümper cohort.

Results:

- Fig. 2h: PPAR γ knock down (actually I believe this is rather a knock out model) of clone 1 was not really efficient, but was not discussed in detail. Please also explain why FABP4 was detected and what you conclude from the results. I could not find anywhere that this was mentioned. However, the results are quite interesting since both KD clones became negative for FABP4 suggesting some crosstalk. It is known that FABP4 can attenuate PPAR γ , but your results suggest also a vice versa regulatory relation? Please comment.
- Fig. 3c: I could not follow why a E:T ratio of 2:1 was used for the empty controls, a ratio which was not used elsewhere. Please comment.
- Supplemental Fig. 4g-h: In the main text you refer to further confirming results using another cell line (HT1197) in animal experiments. However, I found only immunohistochemistry stainings, but no growth curve. Please double check or mark more clear.

Discussion:

- This section lacks a more detailed discussion of the regulatory interaction between NECTIN4, PPAR γ and FABP4 which is the second important aspect of this manuscript apart from CAR T cell generation. Please add.

Methods:

- The source of isolated T cells is not stated. Please do so.
- Co-culture assays are performed for 72-120 h, which I found very long. Killing occurs earlier. Please explain why this interval was chosen. Where the T cells not enough stimulated or active?

Kind regards,

Michèle J. Hoffmann

Reviewer #3

(Remarks to the Author)

In this manuscript titled "Modulating the PPAR γ pathway upregulates NECTIN4 and enhances 2 chimeric antigen receptor (CAR) T cell therapy in bladder cancer", Kevin Chang and co-authors demonstrate that PPAR-gamma regulates the expression of NECTIN4 in bladder cancer cells and therefore the administration of PPAR-gamma antagonists could improve the efficacy of CAR T cells against NECTIN4. In addition, using several human bladder cancer cell lines, the authors also found that up-regulation of NECTIN4 expression by PPAR-gamma antagonists could potentially help overcome the resistance to antibody-drug conjugate enfortumab vedotin.

However, to improve the quality of the manuscript, authors should address several following issues:

1. Fig. 1h and Fig 2h (Western Blotting) demonstrate a high level of Nectin4 expression in the control untreated RT112 cancer line. However, data presented in Fig. 2d and 2b show that the same untreated cells show very low Nectin4 expression. Authors should explain this discrepancy in Western Blotting results.
2. The stimulation of cell surface expression by PPAR-gamma (Fig. 2c) is very low. The authors should provide a quantification for this experiment.
3. Figures 4e and 4j show the in vivo effects of CAR T cell therapy on tumor growth. The authors also should add data on the survival of treated and control mice, showing Kaplan-Meier surviving curves.

Reviewer #4

(Remarks to the Author)

Chang et al. report an interesting study identifying PPAR γ pathway upregulates NECTIN4 and enhances CAR T cell therapy in bladder cancer. The authors demonstrated that NECTIN4 is enriched in luminal subtypes of UC and that its

expression is correlated with multiple luminal transcription factors including GATA3, FOXA1, and PPARG across multiple UC cell lines and clinical samples cohorts. Additionally, PPARG encodes the nuclear receptor peroxisome proliferator-activated receptor γ (PPAR γ), a transcriptional regulator of metabolism whose activity can be modulated via multiple small molecules and by directly binding fatty acids. Authors hypothesized that modulating this pathway may not only regulate NECTIN4 levels, but also enhance NECTIN4-targeting therapies.

Authors developed second-generation CAR T cell candidates with high specificity against NECTIN4 and establish the potency of our CAR T cells in multiple UC cells expressing a range of endogenous NECTIN4. Also they demonstrate that modulating the PPAR γ pathway transcriptionally regulates NECTIN4 expression and leads to increased NECTIN4 protein on the cell surface.

Overall the idea that PPARG pathway upregulates NECTIN4 is not novel, however, identifying therapeutic vulnerabilities in these tumors is important. However, I have major concerns regarding the study design:

1) I refer to upregulates NECTIN4 assessed by IHC following amplification, mutations or translocations. Herein, the authors defined upregulates NECTIN4 tumors among cell lines. In a VERY less number (17) cohort of UC patients treated with EV who underwent pre-EV and post-EV biopsies, NECTIN4 expression was overall higher at the time of EV resistance and was increased in the majority of matched biopsies evaluated (9 of 17). This number is very low.

1a) How the NECTIN4 protein expression and survival association?

1b) How the NECTIN4 mRNA expression and survival association?

1c) In patient cohort is there any correlation of PPARG with NECTIN4 protein expression and survival association? What percentage of patients had this correlation.

2) Transcription factor motif analysis need to perform by ATAC seq (Refer to the recent bladder cancer paper published in Nat Commun. 2024 Feb 14;15(1):1373 PMID: 38355560; to identify the luminal transcription factors including GATA3, FOXA1, and PPARG; transcriptional regulator of metabolism enzymes (based on the author hypothesis) to ensure that directly binding fatty acids/FASN. OR alternatively the authors might comment on the rank of PPARG in HOMER transcription factor analysis.

3) Genetic KD of NECTIN4 had any role in In vivo? Or invitro. Authors need to consider this by utilizing orthotopic UC xenograft model,

4) Figure 5: NECTIN4-CAR T cell therapy demonstrates efficacy in vitro and in vivo against preclinical model of enfortumab vedotin (EV) resistance was done in RT112 cell lines. Authors need to show the evidence in two cell lines.

5) Some of the co culture experiments were performed in different cell lines UMUC-9/umuc3. Authors need to consistent in at least two cell line across the MS.

6) NECTIN4 W. blot had multiple bands, it was hard to believe the quality of the data for NECTIN4 W. blot for entire MS. Authors need to show single band or alternative evidence.

7) Supplementary Figure 4D. How the UMUC1 xenografts were generated? What happened to the GAPDH levels since entire MS shows GAPDH as a controls.

Version 1:

Reviewer comments:

Reviewer #2

(Remarks to the Author)

The authors have addressed all my comments in rebuttal letter and revised version. They also added additional data in response to other reviewer's comments to further improve the manuscript. My recommendation is to accept for publication.

Michèle J. Hoffmann

Reviewer #3

(Remarks to the Author)

Revised version of manuscript is much stronger. All concerns were addressed, no further questions.

Reviewer #4

(Remarks to the Author)

All the comments provided have been carefully reviewed and addressed. After thoroughly assessing the revisions, I find that all concerns and suggestions have been incorporated appropriately. At this stage, I do not have any additional comments or suggestions for further modifications.

I accept the paper.

REVIEWER COMMENTS

Reviewer #1 (Remarks to the Author): with expertise in bladder cancer, immunology

This manuscript by Chang et al aims to evaluate a NECTIN-4 CAR T cell in targeting EV-refractory bladder cancer patients. The main strength of this manuscript is the elegant demonstration of a mechanistic connection between NECTIN-4 transcription and PPAR-gamma, a pathway when induced, known to drive urothelial cell differentiation. They propose that Rosiglitazone-induced NECTIN-4 expression synergizes with NECTIN-4 CAR T cells to enhance its killing, with potential value in EV-resistant patient populations.

1. The major innovation is Figure 2, showing PPAR-g agonists nicely induces NECTIN-4 protein expression while genetic knockout of PPAR-g decrease NECTIN-4 expression; however, it is unclear whether such regulation can be generalized to other high-grade tumor models.

We thank the Reviewer for this comment. We have now evaluated rosiglitazone in multiple breast cancer cells and prostate cancer cells, and we do not find that rosiglitazone is able to induce NECTIN4 expression in these other cancer types. We have included this data below for the Reviewer only (**Reviewer Fig. R1, for reviewer only**).

Reviewer Fig. R1

Reviewer Fig. R1: Rosiglitazone Treatment does not increase NECTIN4 expression in breast and prostate cancer cells. NECTIN4 surface expression in breast MDA-MB-361 (a), MDA-MB-453 (b), T47D (c) and prostate 22Rv1 (d), C42B (e), and LNCaP (f) cells treated with indicated concentrations of rosiglitazone. Western blot of NECTIN4, PPAR γ and Vinculin in indicated breast cancer (g) and prostate cancer (h) cell lines.

2. We commend the authors in generating a specific NECTIN-4 CAR T cells, which is valuable and technically promising. However, Nectin-4 is highly expressed in proximal digestive tract, skin, and bone marrow. It is unclear how these authors plan to circumvent the potential toxicities of NECTIN-4 CAR T cells in human patients beyond this preclinical model?

We thank the Reviewer for this important point. Using RNA-seq data from the GTEX database, we observed that most normal tissues do not co-express *PPARG* and *NECTIN4* (Reviewer Fig R2, for reviewer only, data from the GTEX database), and so we do not believe that increasing *NECTIN4* using rosiglitazone (or other $PPAR\gamma$ agonists) would increase *NECTIN4* levels in normal tissues that would result in unintentional and worsened toxicity. Importantly, we have used multiple Basal bladder cancer cell lines models and were not able to upregulate *NECTIN4* with the addition of rosiglitazone in those cell lines, either because of the lack of $PPAR\gamma$ or *NECTIN4* expression at baseline.

In addition, Dr. John Lee's Lab at UCLA (who is co-author on this manuscript) is currently generating a humanized *NECTIN4* (N4) knock-in (KI) mice, where the mouse *Nectin4* locus is replaced by the human *NECTIN4* locus. We plan to use these mice to assess potential toxicity of our *NECTIN4* CAR T cell therapy in future studies. However, these mice are still being bred (delayed by his move from the Fred Hutch to UCLA, and the quarantine period) and so are not yet available for the current study.

In terms of future translation, given that the *NECTIN4* targeting antibody-drug conjugate enfortumab vedotin is FDA-approved for frontline treatment of locally advanced and metastatic bladder cancer (based on data from Powles et al, *NEJM*, 2024), and that other cyclic binders targeting *NECTIN4* are currently in trial (for example, BT8009 from Bicycle Therapeutics) we believe that there is a therapeutic window to safely administer a *NECTIN4*-targeting CAR T cell therapy as well.

Reviewer Fig. R2

Reviewer Fig. R2: Correlation between *PPARG* and *NECTIN4* mRNA expression in the indicated normal tissues. Luminal subtype bladder cancer is highlighted in the red box.

3. In figure 4e and 4j, it is slightly surprising that the response of RT112 and HT1197 to *NECTIN4* CAR-T alone is comparable when the baseline protein expression of *NECTIN4* seems to be much higher in HT1197 vs RT112. The blots in Fig 4e are run separately than the blots in Fig 4g, and are always internally controlled. These blots cannot be directly compared since they were run on separate days and exposed using ECL detection reagents for different amounts of time. We have to develop blots differently because otherwise, the signal can be over-saturated and not of publication quality. In addition, depending on the basal expression and amount of protein loaded, we may alternate between Pierce Femto vs Pierce Dura ECL developing reagents, which have different sensitivities and need different exposure times.

In addition, because the tumor growth kinetics are different between HT1197 and RT112 xenografts, we do not believe that we can make direct comparisons between the cell lines. From a previous paper, we also showed that HT1197 cells have lower NECTIN4 expression than RT112 cells (see **Reviewer Fig. R3, for reviewer only**, which is taken from Fig. 2D from Chou et al, *Eur Urol Onc*, 2022, PMID: 35216942).

Reviewer Fig. R3

Reviewer Fig. R3: a) Figure adapted Fig. 2D from Chou et al, *Eur. Urol. Onc.*, 2022; PMID: 35216942 demonstrating surface NECTIN4 expression levels in the indicated cell lines. Please note expression levels in RT112 (gray) versus HT-1197 (yellow).

Also, these efficacies are not very encouraging when other TME components, e.g., myeloid are not present. However, it is quite surprising that the response of RT-112-EVRes is quite remarkable when NECTIN-4 protein expression (Figure 5c) is quite marginally increased. Can the authors provide some control experiments or explanation/rationale?

We thank the Reviewer for this comment, and agree that many TME components such as myeloid cells are not present in these animal models. We have now expanded and explicitly mention this in the limitations section of the Discussion:

“Limitations of our current study include the use of immunocompromised mice (which lack some components of the tumor microenvironment (TME) including myeloid cells, which play a role in regulating immunotherapy responses^{67,68}) to evaluate the anti-tumor activity of the CAR T therapy and evaluating tumor responses in non-orthotopic tumor sites.”

We also agree that the RT112-EVres responds very well to the CAR T therapy in Fig 5m. Our main points are that 1) EV resistant cells still express NECTIN4 and 2) EV resistant cells remain sensitive to a different surface protein targeting modality such as a CAR T, despite being resistant to the antibody drug conjugate.

4. In figure 4e and 4j, although rosiglitazone demonstrated added effect when administered together with NECTIN-4 CAR-T. The tumors eventually bounced back. Since NECTIN-4 might be associated with differentiated tumor cells, can be co-stain for stem vs differentiated cell markers in association with NECTIN-4 in the residual tumors?

We thank the Reviewer for this comment. In our model, we only utilized a single dose of CAR T cell injections, and did not repeat CAR T administrations. Also, we found that many of the tumors that did grow in the CAR T treated mice were necrotic or fluid-filled. Therefore, even though a mass was measurable, there wasn't any viable tumor tissue to collect and analyze, and so were limited with the number of samples to perform downstream processing and analyses.

We analyzed the NECTIN4 expression in the tumors that eventually grew out in the NECTIN4 CAR T treated cohort, which show retained NECTIN4 expression, suggesting that re-treatment may be possible. We have included this as **Reviewer Fig. R4a**. We also performed qPCR for multiple luminal and basal/stem cell

markers and do not observe any major differences in these markers. We have included these data in **Reviewer Fig. R4b** and in the manuscript as **Supplementary Fig. 4g-h**.

Reviewer Fig. R4

Reviewer Fig. R4: a) *NECTIN4* expression in tumors isolated from indicated treatment group. Two primer sets for *NECTIN4* were used. b) Expression of indicated luminal and basal markers in tumors isolated from indicated treatment group. N= 4 in Vehicle + NTD T cells; N=3 in Rosiglitazone + NTD T cells; N=3 in Vehicle + N4 CAR T cells; N=4 in Rosiglitazone + N4 CAR T cells. Statistical testing was performed using ANOVA. ns = not significant. ** $P < 0.01$.

Reviewer #2 (Remarks to the Author): with expertise in bladder cancer, immunology

Chang et al. generated CAR T cells targeting NECTIN4 surface protein which is also used for a new targeted therapy of bladder cancer (BC) using the antibody-drug-conjugate (ADC) enfortumab-vedotin (EV; Padcev). This is a smart idea for 2 reasons: 1) EV has an all comer approval by FDA and EMA based on the argument that >80% of bladder cancer tissues are NECTIN4 positive. Thus, CAR T cells recognizing NECTIN4 should be an efficient tool in the majority of BC patients (both muscle-invasive and non-muscle invasive BC). 2) CAR T cells based therapy for solid cancer are still a bit “left behind” due to some difficulties. Thus it is important to further investigate and promote such approaches.

In addition, the authors demonstrate that PPAR γ agonists can augment expression of the target protein. Concurringly, pre-treatment of NECTIN4 low BC could increase target abundance for CAR T and augment treatment efficacy in such cases. This is a sound study using many different control experiments and state of the art methodology and is of high significance for two research communities: 1) those aiming at development of new immunotherapeutic approaches for BC and 2) those trying to move forward CAR T cell based therapeutic approaches for solid cancers. Published data on CAR T cell approaches for BC and on CAR T cells against NECTIN4 is scarce. Thus, I recommend minor revision.

We thank the Reviewer for these positive comments.

My comments are:

- One major points refers to authors’ argument that combined treatment with the PPAR γ agonists and CAR T cells targeting NECTIN4 could be beneficial for EV-refractory patients: 1) However, they mention themselves that a post-EV biopsy cohort retained NECTIN4 expression and also that their EV-resistant cell line model retained the target protein. Following this argument such EC-refractory cases would not have strong need to further increase NECTIN4 levels. In addition, in hour hands killing efficacy of CAR T cells against BC cell lines in vitro was not so much dependent on target density (as mentioned by the authors) and also cell lines with less prominent expression were sufficiently killed. Thus I don’t think that it is “given” that high TAA density is generally needed (introduction and discussion p.8 line 296), please rephrase. Instead I would rather think that addition of a PPAR γ agonist would be beneficial in the about 20% of BC patients with low or no NECTIN4 abundance or those loosing the protein during cancer progression as reported by Klümper et al 2023. This idea would extend suitability of NECTIN4 CAR T therapy to almost all BC patients, which is even better. So maybe you want to reconsider your arguments concerning the EV-refractory setting.

We thank the Reviewer for these comments, and completely agree that there may not be a strong need for tumors that have high NECTIN4 expression to further upregulate expression (for example, likely those with NECTIN4 amplifications). Nonetheless, for tumors with low or even moderate NECTIN4 expression, both in the EV-naive and refractory settings, upregulating NECTIN4 using this approach may be beneficial. It may also help to decrease the number of NECTIN4-targeting CAR T cells needed to be collected and injected (and therefore could be potentially saved for future administrations later on). We agree that the upregulation would be most beneficial in the patients with relatively low NECTIN4 (the 20% or so as suggested by the Reviewer), although the effects of rosiglitazone on the tumor microenvironment warrant future investigation study as well. We have revised the Discussion as suggested by the Reviewer:

“In this study, we also show that upregulating NECTIN4 using rosiglitazone (a PPAR \$\gamma\$ agonist, originally developed for the treatment of diabetes), can enhance the efficacy of the NECTIN4 CAR T cell therapy, which is particularly important for tumors with low or moderate NECTIN4 expression, both in the EV-naïve and EV-refractory settings, which may improve the durability of response. This is consistent with previous reports showing that CAR T cell efficacy is related to the level of TAA expression on tumor cells^{13,47}. Our strategy of repurposing rosiglitazone to prime NECTIN4 expression across a population of tumor cells with low to moderate expression significantly enhanced anti-tumor activity of NECTIN4-targeting CAR T therapy.”

- Please also discuss further retained NECTIN4 in the mentioned post EV cohort and reduced expression in the Klümper cohort.

We thank the Reviewer for this comment. To our understanding, the Klumper cohort (Klumper et al, *Clin Cancer Res*, 2023, PMID: 36534531) did not analyze any post-EV biopsies, but rather described biopsies taken from primary tumors and matched patients who also had metastatic biopsies. These patients may have been treated with or without any therapy in the interim. In their cohort, Klumper and colleagues found reduced expression in the metastatic biopsies compared to the primary tumor. Our biopsy cohort is different in 2 important ways: 1) it is a pre/post EV cohort, and 2) nearly all the biopsies are from metastatic sites, so we are comparing metastatic biopsy to metastatic biopsy. Our staining methods may also be slightly different, but they are all internally controlled.

Results:

- Fig. 2h: PPAR γ knock down (actually I believe this is rather a knock out model) of clone 1 was not really efficient, but was not discussed in detail. Please also explain why FABP4 was detected and what you conclude from the results. I could not find anywhere that this was mentioned. However, the results are quite interesting since both KD clones became negative for FABP4 suggesting some crosstalk. It is known that FABP4 can attenuate PPAR γ , but your results suggest also a vice versa regulatory relation? Please comment.

We agree that guide 1 was less efficient, and that because the degree knockdown varies from sgRNA to sgRNA, we used two independent sgRNA sequences. Furthermore, we utilized HPGD and FABP4 previously established downstream targets and indicators of PPAR γ pathway activity (Bahrami-Nejad et al, *PLoS Biol*, 2022; PMID: 36469503), and FABP4 as a control and reasoned that knocking down PPAR γ would decrease FABP4. While there are studies demonstrating that FABP4 can attenuate PPAR γ (Bervejillo et al, *Redox Biol*, 2019; PMID: 31926616), the only purpose of our blot is to show that knocking down PPAR γ inhibits its downstream target, FABP4. (We can remove this blot from Fig 2h if the Reviewer thinks it is too confusing.)

- Fig. 3c: I could not follow why a E:T ratio of 2:1 was used for the empty controls, a ratio which was not used elsewhere. Please comment.

We apologize for the discrepancy. We had initially plotted the E:T ratio of 2:1 (which includes more T cells) to show that these Empty-CAR or Non-transduced T cells are unable to kill tumor cells, skewing the experiment *against* our favor to evaluate if there was any non-specific, allogeneic killing. The NECTIN4 CAR T cells exhibit anti-tumor activity at lower E:T ratios. To minimize the inconsistencies across the panels, we have now plotted Fig. 3c with an E:T of 1:1.

- Supplemental Fig. 4g-h: In the main text you refer to further confirming results using another cell line (HT1197) in animal experiments. However, I found only immunohistochemistry stainings, but no growth curve. Please double check or mark more clear.

We apologize for the confusion. The growth curves for the HT1197 xenograft model are included in the main Fig 4k-l, and this has been amended in the main text to be more clear.

Discussion:

- This section lacks a more detailed discussion of the regulatory interaction between NECTIN4, PPAR γ and FABP4 which is the second important aspect of this manuscript apart from CAR T cell generation. Please add. We have added a sentence and reference describing the previously published work outlining the relationship between NECTIN4, PPAR γ and FABP4. As previously mentioned above, we used FABP4 as a downstream target of PPAR γ to confirm whether we were able to activate/suppress PPAR γ .

“We found that in UC cells, PPAR γ regulates *NECTIN4*, as well as other previously described downstream genes such as *FABP4*⁵¹.”

Methods:

- The source of isolated T cells is not stated. Please do so.

Our T cells are isolated from a healthy human donor, peripheral blood Leukopak from Stem Cell Technologies, catalog #70500.2. We have now added this in the Methods section.

- Co-culture assays are performed for 72-120 h, which I found very long. Killing occurs earlier. Please explain why this interval was chosen. Where the T cells not enough stimulated or active?

For the majority of the co-culture assays, we performed the co-culture until 72 hours, and found that most killing occurred within 48 hours, particularly in wells with higher E:T ratios. However, for the HT1197 cells, we did extend the experiment out to 120 hours in the Incucyte because the T cell killing seemed to take longer (perhaps because the HT1197 cells are bigger, and the T cells seemed to take longer to be activated). Since we obtained data from all of these timepoints, we have shown all the data out to 120h for Fig. 3e for completeness.

Reviewer #3 (Remarks to the Author): with expertise in bladder cancer, immunology

In this manuscript titled “Modulating the PPAR γ pathway upregulates NECTIN4 and enhances 2 chimeric antigen receptor (CAR) T cell therapy in bladder cancer”, Kevin Chang and co-authors demonstrate that PPAR-gamma regulates the expression of NECTIN4 in bladder cancer cells and therefore the administration of PPAR-gamma antagonists could improve the efficacy of CAR T cells against NECTIN4. In addition, using several human bladder cancer cell lines, the authors also found that up-regulation of NECTIN4 expression by PPAR-gamma antagonists could potentially help overcome the resistance to antibody-drug conjugate enfortumab vedotin.

However, to improve the quality of the manuscript, authors should address several following issues:

We thank the Reviewer for these positive comments.

1. Fig. 1h and Fig 2h (Western Blotting) demonstrate a high level of Nectin4 expression in the control untreated RT112 cancer line. However, data presented in Fig. 2d and 2b show that the same untreated cells show very low Nectin4 expression. Authors should explain this discrepancy in Western Blotting results.

The discrepancy is due to running separate blots and obtaining different exposure times for the different figure panels. For all of our blots, the lanes are normalized internally. Exposure times are different between Fig. 1h and Fig. 2h (longer exposure) and Fig. 2b and Fig. 2d (shorter exposure), and so should not be cross-compared since these are different blots. (If Fig. 2b/2d had not had a shorter exposure time, then the blot would be over-saturated and the bands would coalesce together). For panels where all cell lines are blotted together (for example, Fig 1h and Supplementary Fig 1m), then the levels can be compared across cancer cell lines.

2. The stimulation of cell surface expression by PPAR-gamma (Fig. 2c) is very low. The authors should provide a quantification for this experiment.

We thank the Reviewer for this comment. We have quantified the median fluorescence intensity (MFI) (Reviewer Fig. R5a-b), and included it as a new part of Fig. 2c-2d.

Reviewer Fig. R5: a) Surface NECTIN4 expression levels in RT112 cells treated with DMSO control (black), rosiglitazone (orange), or T0070907 (red). b) Quantification of NECTIN4 median fluorescence intensity (MFI) in RT112 cells treated with indicated conditions. N=4 biologically independent experiments, with each data point shown. ANOVA with multiple testing correction (Sidak) was used to compare groups, and P values are indicated.

3. Figures 4e and 4j show the in vivo effects of CAR T cell therapy on tumor growth. The authors also should add data on the survival of treated and control mice, showing Kaplan-Meier surviving curves.

We have now added the Kaplan-Meier graphs for this experiment as suggested by the Reviewer (Reviewer Fig. R6a-b) and have added these data as new Fig. 4f and Supplementary Fig. 4h. We also show the median overall survival of each cohort in the figure.

Reviewer Fig. R6

Reviewer Fig. R6: **a)** Kaplan-Meier survival curve of mice bearing RT112 xenografts treated in the indicated groups using donor 1 NECTIN4 CAR T cells. Log-rank test $p = 0.00018$. **b)** Kaplan-Meier survival curve of mice bearing RT112 xenografts treated in the indicated groups using donor 2 NECTIN4 CAR T cells. Log-rank test $p = 0.0005$.

Reviewer #4 (Remarks to the Author): with expertise in bladder cancer

Chang et al. report an interesting study identifying PPAR γ pathway upregulates NECTIN4 and enhances CAR T cell therapy in bladder cancer. The authors demonstrated that NECTIN4 is enriched in luminal subtypes of UC and that its expression is correlated with multiple luminal transcription factors including GATA3, FOXA1, and PPARG across multiple UC cell lines and clinical samples cohorts. Additionally, PPARG encodes the nuclear receptor peroxisome proliferator-activated receptor γ (PPAR γ), a transcriptional regulator of metabolism whose activity can be modulated via multiple small molecules and by directly binding fatty acids. Authors hypothesized that modulating this pathway may not only regulate NECTIN4 levels, but also enhance NECTIN4-targeting therapies.

Authors developed second-generation CAR T cell candidates with high specificity against NECTIN4 and establish the potency of our CAR T cells in multiple UC cells expressing a range of endogenous NECTIN4. Also they demonstrate that modulating the PPAR γ pathway transcriptionally regulates NECTIN4 expression and leads to increased NECTIN4 protein on the cell surface.

Overall the idea that PPAR γ pathway upregulates NECTIN4 is not novel, however, identifying therapeutic vulnerabilities in these tumors is important. However, I have major concerns regarding the study design:

1) I refer to upregulates NECTIN4 assessed by IHC following amplification, mutations or translocations. Herein, the authors defined upregulates NECTIN4 tumors among cell lines. In a VERY less number (17) cohort of UC patients treated with EV who underwent pre-EV and post-EV biopsies, NECTIN4 expression was overall higher at the time of EV resistance and was increased in the majority of matched biopsies evaluated (9 of 17). This number is very low.

We thank the Reviewer for this comment. Although the number of biopsies described is not very high, obtaining metastatic biopsies for bladder cancer patients is very challenging. The nodes are often in locations that are impossible to reach with a needle (retroperitoneal, pelvic nodes), and many patients do not provide consent to obtaining these research biopsies (for multiple reasons, including not feeling well while on therapy, and the potential risks of infection, bleeding, etc.). These are all also considered research biopsies, and not standard of care biopsies. To the best of our knowledge, our cohort of 17 patients is the largest pre/post EV biopsy set described to date.

1a) How the NECTIN4 protein expression and survival association?

We previously showed that *NECTIN4* expression is not associated with survival in multiple cohorts that we examined (Chu et al, *Clin Cancer Res*, 2019, PMID: 34108177; Supplementary Figure 2E from that paper is provided below, **Reviewer Fig. R7, for reviewer only**). In addition, recent data from the 2024 ESMO conference presented by Dr. Thomas Powles showed that patients with high vs low NECTIN4 H-scores had similar survivals when treated with chemotherapy. (We can provide the PDF of that presentation if requested; to our knowledge, that data has not yet been published in a peer-reviewed manuscript).

Reviewer Fig. R7

Reviewer Fig. R7: Figure adapted from Chu et al, *Clin Can Res*, 2019; PMID: 34108177. Kaplan-Meier survival curves of overall survival stratified by NECTIN4 expression in the TCGA dataset (p=0.11), Sjordahl 2012 dataset (p=0.6) and Seiler 2017 dataset (p=0.06).

1b) How the NECTIN4 mRNA expression and survival association? –

We previously showed that *NECTIN4* expression is not associated with survival (Chu et al, *Clin Cancer Res*, 2019, PMID: 34108177, please see **Reviewer Fig. R7** above).

1c) In patient cohort is there any correlation of PPARγ with NECTIN4 protein expression and survival association? What percentage of patients had this correlation.

We previously showed that NECTIN4 expression is associated with PPARγ expression (Chu et al, *Clin Cancer Res*, 2019, PMID: 34108177– see below which is adapted from our Fig. 1D, which included n=529 patient samples) in multiple patient cohorts (**Reviewer Fig. R8a, for reviewer only**). In addition, we have evaluated a separate cohort of patients in which we obtained pre-EV biopsies only, and found an association between

NECTIN4 and PPAR γ H-scores (Pearson correlation coefficient 0.646, $p = 0.0006$, $n=24$ patients) (Reviewer Fig. R8b, for reviewer only).

Reviewer Fig. R8

Reviewer Fig. R8: a) Figure adapted from Chu et al, *Clin Can Res*, 2019; PMID: 34108177 demonstrating the correlation between *PPARG* and *NECTIN4* expression. The Pearson correlation coefficient (0.43) and p value (3.8e-45) are indicated. b) Correlation between *NECTIN4* and PPAR γ H-scores, with the Pearson correlation coefficient (0.646) and p value (0.0006) are indicated.

2) Transcription factor motif analysis need to perform by ATAC seq (Refer to the recent bladder cancer paper published in Nat Commun. 2024 Feb 14;15(1):1373 PMID: 38355560; to identify the luminal transcription factors including GATA3, FOXA1, and PPAR γ ; transcriptional regulator of metabolism enzymes (based on the author hypothesis) to ensure that directly binding fatty acids/FASN. OR alternatively the authors might comment on the rank of PPAR γ in HOMER transcription factor analysis.

This is an interesting point raised by the reviewer. However, we did not perform ATAC-seq in our study, and although would be extremely interesting, believe this is beyond the scope of our study. In addition, we do not make any claims about direct binding of fatty acids/FASN in our study, although we did show that arachidonic acid does increase NECTIN4 levels (Supplementary Fig. 2g-h). This is certainly an interesting point and studies on how dietary fat might affect NECTIN4 in bladder cancer patients warrant future investigation, but is outside the scope of this current manuscript. Regarding the rank of PPAR γ , we used FIMO (version 5.4.1 compiled on Oct 8, 2023) to assess for potential transcription factor binding sites in the promoter region, which we defined at -2kb upstream of the transcriptional start site. The PPAR γ site we found and evaluated ranked #17 (among 579 binding sites) with a significant p value of 2E-6. In addition, other PPAR γ and/or RXR binding sites were found within the promoter region of *NECTIN4*.

3) Genetic KD of NECTIN4 had any role in In vivo? Or in vitro. Authors need to consider this by utilizing orthotopic UC xenograft model.

We evaluated the cell proliferation kinetics of RT112 Control and NECTIN4 KO cell lines, and found no differences in proliferation between the cell lines *in vitro* (Reviewer Fig. R9a, now added as Supp Fig 1c) or tumor growth *in vivo* (Reviewer Fig. R9b, for reviewer only). We also assessed NECTIN4 overexpression (OE) in the UMUC3 cells and found that there was no difference in tumor growth (Reviewer Fig. R9c, for reviewer only). Taken together, these data suggest that NECTIN4 has no effect on proliferation *in vitro* or tumor growth *in vivo*, at least in these models that were tested. Future studies will be aimed towards understanding NECTIN4's biological function, which we believe are beyond the scope of the current study.

Reviewer Fig. R9

Reviewer Fig. R9: **a)** Cell proliferation assay of RT112 Parental and 2 *NECTIN4* knockout (KO) clones. A representative experiment performed in triplicate wells of n=3 biologically independent experiments is shown. **b)** Tumor xenograft growth of RT112 Control (n=4 mice) and RT112 *NECTIN4* KO1 (n=6 mice) tumors. **c)** Tumor xenograft growth of UMUC3 Control (n=10 mice) and UMUC3 *NECTIN4* overexpressing (OE) tumors (n=8 mice). Error bars represent the standard error of the mean (SEM).

4) Figure 5: *NECTIN4*-CAR T cell therapy demonstrates efficacy in vitro and in vivo against preclinical model of enfortumab vedotin (EV) resistance was done in RT112 cell lines. Authors need to show the evidence in two cell lines.

We thank the Reviewer for this comment. We have now added data on a second EV resistant cell line, the UMUC1 EV RES line, which has an IC₅₀ of 6.8 ug/ml (compared to an IC₅₀ of 0.23 ug/ml in the parental UMUC1 cells) (**Reviewer Fig. R10a**). We found that the *NECTIN4* CAR T therapy retains potent anti-tumor activity against this second EV resistant cell line, which has similar to slightly lower *NECTIN4* levels (**Reviewer Fig. R10b-d**). We have added these to the Supplementary data (**new Supplementary Fig. 6**).

Reviewer Fig. R10

Reviewer Fig. R10: **a)** EV dose-response curves for UMUC-1 Parental and EV-resistant (RES1) cells. The IC₅₀ in parental and EV RES1 cells is 0.23 ug/ml and 6.8 ug/ml, respectively. **b)** Western blot showing *NECTIN4* levels in the UMUC1 Parental and EV RES1 whole cell lysates. Vinculin is used as the protein loading control. **c)** Surface *NECTIN4* expression in the UMUC-1 Parental and UMUC-1 EV RES1 cells. **d-e)** Representative curves of UMUC-1 Parental and UMUC1 EV RES1 cells co-cultured with *NECTIN4*-CAR T cells (d) or NTD T cells (e) at an E:T of 1:1. For panels a, d, and e, the error bars represent standard error of the mean, and a representative experiment of n=3 biologically independent experiments performed in technical triplicates is shown.

5) Some of the co culture experiments were performed in different cell lines UMUC-9/umuc3. Authors need to consistent in at least two cell line across the MS.

We have used RT112 and UMUC-1 cell lines consistently across the manuscript and figures. For many experiments, we also validated our findings in a 3rd or 4th cell line, for example, HT1197 and UMUC-9 cells, to

demonstrate the generalizability of our findings. However, one instance in which we did not use the UMUC-1 cells was in generating the NECTIN4 KO. This is because UMUC-1 cells have low endogenous levels of NECTIN4. Therefore, for the NECTIN4 knockout experiments, we chose to use a 2nd line with higher basal expression, (i.e., UMUC-9) to demonstrate that loss of NECTIN4 abolishes the NECTIN4 CAR T activity.

For the NECTIN4 overexpression studies, we chose to perform these experiments in the UMUC-3 cells, which have no detectable endogenous NECTIN4 expression (i.e., the expression is completely negative by western blot and flow cytometry). This was intentional, because these cells have no endogenous NECTIN4 expression, to study the effects of expressing NECTIN4 on CAR T kill with a clean negative background. However, to address the Reviewer’s concern, we have now also generated UMUC1-NECTIN4 over-expressing (OE), and demonstrate that upregulating NECTIN4 enhances NECTIN4 CAR T cell killing in the UMUC-1 cells as well. These data are shown below (**Reviewer Fig. R11a-b**) and we added these to the Supplementary data (new **Supplementary Fig 1j-k**)

Reviewer Fig. R11

Reviewer Fig. R11: **a)** NECTIN4 surface protein expression in UMUC-1 (blue) and UMUC-1 N4 overexpression (OE) cells. The negative unstained population is shown in gray as a control. **b)** Growth curves of UMUC-1 parental (blue) and UMUC-1 NECTIN4 OE (red) target cells co-cultured with NECTIN4-CAR T cells at an E:T ratio of 1:1. Error bars represent standard error of the mean. A representative experiment of n=3 biologically independent experiments performed in technical triplicates is shown.

6) NECTIN4 W. blot had multiple bands, it was hard to believe the quality of the data for NECTIN4 W. blot for entire MS. Authors need to show single band or alternative evidence.

NECTIN4, like all surface proteins, is a heavily glycosylated protein. (REF: <https://www.ncbi.nlm.nih.gov/pmc/articles/PMC10196828/>), and so therefore, there are multiple bands when detecting this protein on western blot. Below, we have included the representative blot images from Abcam, Cell Signaling Technology and ThermoFisher: this includes both monoclonal (Abcam #ab189514 and ab192033) and polyclonal (CST #17402, ThermoFisher 21903-1-AP) antibodies (**Reviewer Fig. R12, for reviewer only**). For all of the antibodies, there are multiple bands detected, including the glycosylated forms, for every product. The size of NECTIN4 by western blotting ranges from approximately 37-65 kDa, per every manufacturer.

In terms of alternative evidence, we completely agree that this is important, and so we have also shown NECTIN4 levels using a fluorescently-labeled antibody from Miltenyi throughout the manuscript by flow cytometry (for example, **Figures 1c, 2c, 2f, 5d**). We have also internally validated levels using as the antibody from Thermo (21903-1-AP). Both the Miltenyi and Thermo antibodies can detect surface NECTIN4. Therefore, we believe that the combination of multiple antibodies and validation strategies confirms our western blotting data.

Reviewer Fig. R10

Reviewer Fig. R8: a-e) Representative western blots taken from the indicated manufacturer's website demonstrating the glycosylated NECTIN4 forms for commercially available NECTIN4 antibodies from abcam (a-c), Cell Signaling Technology (d) and ThermoFisher (e). These antibodies are commonly published on by other researchers in the field as well as our group previously (Chu et al, *Clin Can Res*, 2020).

7) Supplemental Figure 4D. How the UMUC1 xenografts were generated? What happened to the GAPDH levels since entire MS shows GAPDH as a controls.

These tumors from Supp. Fig 4D are subcutaneous UMUC-1 xenografts. We use both GAPDH (MW 37 kD) and vinculin (MW 124 kD) consistently as loading controls throughout the manuscript. To minimize stripping blots and re-probing, we typically tried to choose a loading control that had a different MW than other proteins of interest. For example, because HPGD is 24 kD and GAPDH is 37 kD, when we were probing for HPGD, then we used vinculin (124 kD) to avoid overlapping bands. Therefore, for the tumor xenograft blots, we used vinculin as the loading control since we also probed for HPGD to verify that the mice successfully received the rosiglitazone treatment.

NCOMMS-24-40264A: Chang et al

REVIEWERS' COMMENTS

Reviewer #2 (Remarks to the Author):

The authors have addressed all my comments in rebuttal letter and revised version. They also added additional data in response to other reviewer's comments to further improve the manuscript. My recommendation is to accept for publication.

Michèle J. Hoffmann

Thank you for the positive comments and review.

Reviewer #3 (Remarks to the Author):

Revised version of manuscript is much stronger. All concerns were addressed, no further questions.

Thank you for the positive comments and review.

Reviewer #4 (Remarks to the Author):

All the comments provided have been carefully reviewed and addressed. After thoroughly assessing the revisions, I find that all concerns and suggestions have been incorporated appropriately. At this stage, I do not have any additional comments or suggestions for further modifications. I accept the paper.

Thank you for the positive comments and review.